# Activities of daily living, self-efficacy and motor skill related fitness and the interrelation in children with moderate and severe Developmental Coordination Disorder

**Faiçal Farhat**[1]*, **Marisja Denysschen**[2], **Nourhen Mezghani**[3], **Mohamed Moncef Kammoun**[1], **Adnene Gharbi**[4], **Haithem Rebai**[5], **Wassim Moalla**[1], **Bouwien Smits-Engelsman**[2]

**1** Research Laboratory: Education, Motricity, Sport and Health, EM2S, LR19JS01, High Institute of Sport and Physical Education of Sfax, University of Sfax, Sfax, Tunisia, **2** Physical Activity, Sport and Recreation, Faculty Health Sciences, North-West University, Potchefstroom, South Africa, **3** Department of Sport Sciences, College of Education, Taif University, Taif, Saudi Arabia, **4** Physical activity, Sport and Health Research Unit, National Observatory of Sport, Tunis, Tunisia, **5** Sports Performance Optimization Research Laboratory (LR09SEP01), National Center for Sports Medicine and Science (CNMSS), Tunis, Tunisia

\* faicalfarhat@gmail.com

**Data Availability Statement:** All relevant data are within the paper and its Supporting Information files.

## Abstract

### Background

Children with Developmental Coordination Disorder (DCD) are diagnosed based on poor motor skills that impact their daily activities. However, this may also lead to lower predilection and participation in physical activities and a higher risk to develop health problems.

### Objective

To determine motor skill related levels in children with moderate and severe DCD and compare that to typically developing children (TD). The study also aimed to determine the level of activities of daily living (ADL) as reported by their parent and self-efficacy as reported by the children. Lastly, the relation of motor skill related fitness, ADL, and self-efficacy has been examined.

### Methods

A cross-sectional study has compared TD children (n = 105) and children with DCD (n = 109; 45 moderate DCD and 64 severe DCD) from elementary schools in Tunisia between 7 and 10 years of age. The DCDDaily-Questionnaire and Children's Self-perceptions of Adequacy in and Predilection for Physical Activity Questionnaire have been used to determine ADL and adequacy towards physical activity, respectively. The PERF-FIT has been used to measure motor skill related fitness levels.

**Funding:** Funding source: The Ministry of Higher Education and Scientific Research Tunisia The funders had no role in study design, data collection and analysis, decision to publish, or preparation of the manuscript.

**Competing interests:** The authors have declared that no competing interests exist

## Results

Large group differences (p = 0.001) were found for ADL and all domains of self-efficacy between TD and DCD children. However, ADL and self-efficacy were not different between moderate and severe DCD. Motor skill related fitness was significantly lower (p = 0.01) for children with DCD than TD children, and between children with moderate and severe DCD. The relation between self-efficacy and DCDDaily-Q was different in the two DCD groups. Slow motor learning was associated with lower perceived enjoyment in physical education in the moderate DCD group, and with lower perceived adequacy in physical education in the severe DCD group.

## Conclusions

Children with DCD participate and enjoy physical activity less than their peers. This combination of lower participation, lower predilection to physical activity and lowered motor skill-related fitness is a concern for the present and future health status of children with DCD.

## Introduction

Developmental coordination disorder (DCD) is a pervasive neurodevelopmental disorder, which affects the child's ability to perform everyday motor skills (dressing, playing, writing, hopping, catching, etc.). With a prevalence of 5%, DCD is one of the most prevailing disorders of motor control and motor learning in children [1].

Although the core symptoms in DCD are motor, many other domains are implicated. These include executive function, social and emotional difficulties [2]. Furthermore, there is a common co-occurrence with other disorders such as autism spectrum disorder and attention deficit/hyperactivity disorder [3]. Children with these complex developmental needs are at greater risk of inactivity and reduced fitness levels [4].

What is important, for the topic of this paper, is the restricted participation in active play and organized sports that has been reported [5, 6]. Both motor skill competency and fitness are critical for participation in active play [7]. Compared to typically developing (TD) children, children with DCD are particularly likely to exhibit reduced fitness levels [8, 9]. The two types of physical fitness, most often identified, are health-related physical fitness and motor skill-related physical fitness [10]. Health-related physical fitness consists of those components of physical fitness that have a relationship with good health. The components are commonly defined as body composition, cardiovascular fitness, flexibility, muscular endurance, and strength. Motor skill-related fitness incorporates agility, balance, coordination, speed, power, and reaction time, reflecting the performance aspect of physical fitness [11]. Some studies suggest that motor performance, physical activity, and physical fitness influence each other [9, 10]. Notably, poor motor functioning discourages children from participating in physical activities, that's why they are more likely to be overweight and obese than TD children [12, 13].

By definition, children with DCD have poor motor skills, so it is understandable that they do not feel adequate for physical fitness tasks [15]. They perceive themselves to be less competent in basic physical skills as well as in their overall physical abilities when compared to TD children [14]. Children's perception of their abilities can also have an influence on test results, because they may give up earlier or perceive they are fatigued sooner. Children with DCD experience considerable difficulties to control their body movements during functional motor

tasks [12]. In some tests, like for anaerobic muscle endurance (e.g. the sit- or push-ups executed in 30 s), the poor coordination of repetitive movements of the tasks as well as perceptions of poor physical fitness can have a negative impact on performance. In this context, Le et al. [15] confirmed that it is important to evaluate the perceptions of motor ability as they mediate the relationship between motor difficulties and performance using the standing long jump test. Our current study extends on Cairney's work [16], who reported that children with DCD had lower levels of generalized self-efficacy regarding physical activity than TD children. The impact of poor motor skills on self-perceptions of competence, predilection toward and enjoyment of physical activity has important repercussions for interventions, whether for motor skills training or achieving fitness goals [17].

Children with DCD face evident motor difficulties in daily functioning as gross motor activities, eating, maintaining personal hygiene and dressing. These motor difficulties are debilitating in everyday tasks (criterion B), (APA (DSM-5). However, little is known, about the children's difficulties in specific activities of daily living (ADL) and participation in ADL [18]. Moreover, the scarce evidence available comes from small samples without taking into account the severity of the motor skill impairment. Poor motor coordination will hamper many ADL activities, keeping balance while putting on pants, close shirt buttons, lace shoes or drink from a cup without spilling. Information about the impact of the severity of motor coordination level on mastering these skills will improve the understanding of the disorder and help to define priorities for rehabilitation. According to the international guideline for DCD [19], standardized questionnaires are advised for systematically assessing the burden of poor motor coordination on ADL. The most frequently used questionnaire worldwide is the DCD-Q [20]. A more elaborate standardized assessment tool of ADL for children with DCD, used in this study, is the DCDDaily-Q which evaluates not only ADL performance, but also the time it takes to learn the task and how often children participate in the task. Van der Linde et al. [20] reported that children with DCD had lowered performance in ADL, delays in learning of ADL and participated less frequently in ADL than TD children [20]. Importantly, Ferguson et al. [12] showed that tasks that should have been learned implicitly during play and everyday activities (running up and down stairs, lifting a box, sit to stand from a chair) were consistently poor in children with DCD. Additionally, adequate muscle strength and endurance are important for performing many of these daily activities and to participate in sports without early fatigue [21].

Because of their difficulties with the coordination of fine and gross motor skills, children with DCD are usually unable to successfully participate in school, sport and leisure games, thus studying relationships among these aspects is important.

Collectively, the literature shows that children with DCD participate less in physical activities compared to their peers, especially in team sports [19]. It is therefore critical to identify children who lack the fitness prerequisites needed for sports activities at a younger age. To make the picture complete, reports of what parents see their children actually do during everyday life and the perspective of the child are needed in addition to standardized motor and fitness tests as that information and their mutual relation will be the starting point for any intervention.

Although, low values for motor skill-related fitness components, participation and perceived competency have been noted in the literature. Studies having data on all these components in a large group of children with verified DCD criteria mentioned in the Diagnostic and Statistical Manual of Mental Disorders, fifth edition (DSM-5), over the full range of low motor scores and from a non-Western background are scarce. Little work has examined the consequences of the level of motor deficiency regarding these issues. Therefore, we included two

groups of children with DCD, one with significant movement difficulties and one at risk for motor difficulties.

In this study, we addressed four research questions:

1. Is motor skill-related fitness different in the three groups of children (severe DCD, moderate DCD and TD)?

2. Do parents rate activities of daily living different for the three groups of children?

3. Is the self-efficacy reported by the three groups of children different?

4. How high is the association between the measured motor skill-related fitness, the perceived level of ADL by the parent and the child's perception of self-efficacy?

Motor skill-related fitness, activities of daily living and self-efficacy are expected to be the lowest in the s-DCD group and best in the typically developing group. In line with the summarized literature, we expect that longer learning times of ADL and lower participation in ADL, as well as lower levels of generalized efficacy will be associated with poor motor skills. Low to moderate correlations are expected between actual motor skill-related fitness levels, perceived level of ADL by the parent and the child's perception of self-efficacy in the DCD group.

## Materials and methods

### Study design and procedure

The protocol was approved by the Ministry of Education and the Ethical Committee at the University Hospital Sfax, Tunisia (CPP SUD N° 0301/2021), in accordance with the Code of Ethics of the World Medical Association (Declaration of Helsinki). This cross-sectional investigation included students in grades 3 to 5, from 6 elementary schools in Tunisia. The recruitment of participants lasted 3 months from February 1, 2021 until April 30, 2021. Written informed consent was obtained from all parents. The consent has been verified and approved by the ethics commission. The backward-forward translation method [22] was used to translate the DCDDaily-Questionnaire (DCDDaily-Q) and the Children's Self-perceptions of Adequacy in and Predilection for Physical Activity (CSAPPA) from English to Arabic. As Tunisian children were familiar with the activities and games included in the DCDDaily-Q, items were not changed. The DCDDaily-Q was sent to the parents, who returned them after completion. The CSAPPA was individually administered to the children. The physical education teacher explained the questions. The children should choose the option that best describes them from pairs of statements such as "Some kids like to play with computers". and "Other kids don't like playing with computers." by indicating whether the selected sentence was either "sort of true for me" or "really true for me". Six research assistants, trained by an experienced physiotherapist, conducted the Movement Assessment Battery for Children, Second Edition (MABC-2) and the Performance and Fitness (PERF-FIT) battery in a separate room in the schools. Each child received verbal instructions, demonstrations, and practice trials before the test. BMI was determined from height and weight measurements taken by trained research assistants.

### Participants

Only children who met all the criteria of DSM-5 for DCD [1] were included in the DCD group. To select these participants, teachers and parents were asked to identify children with motor coordination problems based on their observations on the playground, in class or at home. Among children who were identified as having a motor coordination problem by the parent or teacher, the DCDDaily-Q was used to further assess interference of the motor

impairment with daily activities and/or academic achievement (Criterion B). If these children scored at or below the 16[th] percentile on the MABC-2 (Criterion A), and their parents reported that these problems were noticed at a young age (Criterion C), and if they had no reported diagnosis of a significant medical condition or comorbidity known to affect motor performance as mentioned in the parental questionnaire (Criterion D) [20]; and if their teacher affirmed the absence of intellectual or cognitive impairment (Criterion D), these children appeared to fulfill the criteria for DCD. TD children were recruited from the same classes of the school as the children with DCD. Criteria for the TD children: 1) no evidence of functional motor problems as observed by their teacher or parent, 2) a score above the 16[th] percentile on the MABC-2, 3) no diagnosis of a significant medical condition as reported by a parent and 4) absence of intellectual or cognitive impairment as confirmed by their teacher. The study sample size was calculated through a power analysis which showed that a total sample size of 90 per group was needed for a medium effect size (d = 0.5), at a power of 90%, while alpha is set at 0.05 with an allocation ratio of 1. The G-power analysis software version 3.1 was used for the sample size calculation [23]. In total 141 children were identified meeting the diagnostic criteria for DCD out of 2170 children selected from the 6 elementary schools. Typically developing children were matched according to age and sex to the children with DCD. Finely, 214 children performed all the tests, and participated in the study: 109 children with DCD and 105 typically developing matched controls. The recruitment steps are depicted in Fig 1.

## Measurements

**MABC-2.** The MABC-2 test consists of eight items that are evaluated in three different components: manual dexterity, aiming and catching, and balance. Age band 2 of the MABC-2 was used to classify the children into 3 groups. Percentile scores of five or less indicate severe

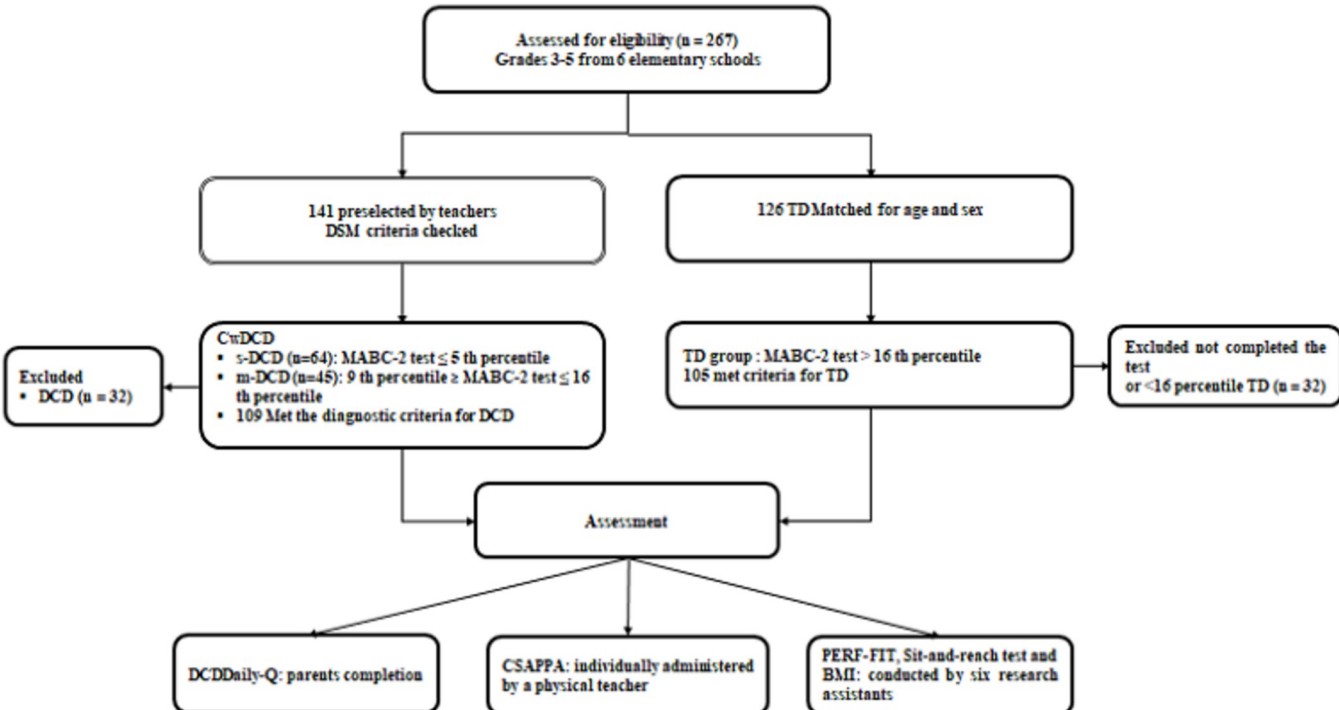

**Fig 1. Flowchart of the selection of children and tests that were administered.** DCD = Developmental Coordination Disorder, TD = Typically Developing, m-DCD = Moderate DCD, s-DCD = Severe DCD.

motor problems (we will refer to this group as severe DCD or s-DCD), while a score between 9 and 16 suggests that the child is at risk of having movement difficulties (we will refer to this group as moderate DCD or m-DCD) and a score >16th percentile indicates normal motor performance [24, 25].

**PERF-FIT battery.** The PERF-FIT is a functional measure of motor skill-related fitness for children [26]. The PERF-FIT comprises two subscales: Motor performance and Power and Agility. The motor performance subscale, incorporates the motor potential to carry out a physical activity, consists of skill item series for bouncing and catching, throwing and catching, jumping and hopping, static and dynamic balance. The power and agility subscale consist of five test items; running (agility), stepping (agility), side jump (muscular endurance), the standing long jump (muscular power of the legs) and the overhead throw (muscular power of the arms). Total performance and subscales levels were calculated according to preliminary age and gender based African norms [26]. The PERF-FIT battery is a valid and reliable test for children aged 5–12 years, with excellent content validity (content validity index ranging from 0.86 to 1.00), good structural validity [27], excellent inter-rater reliability (ICC. 0.99), good test-retest reliability (ICC. ≥ 0.80) [28]. A sit-and-reach test was added as a measure of flexibility and was assessed using a standardized wooden box. Participants are required to sit with knees straight and legs together and the soles of their feet against a box. They had to bend forward and reach as far as possible, with two hands on top of each other and hold the position for at least 3 s. The best score out of 3 trials (in cm) was further analyzed. The level of the feet is used as zero, so that if children did not reach their toes the value is negative and if they reached past their toes the value is positive. The sit and reach test has the advantage of allowing for a simple estimation of hamstrings flexibility and lower back in a short amount of time [29]. The PERF-FIT was used to answer research question 1.

**DCDDaily-Q.** The DCDDaily-Q contains 23 items which are subdivided in self-care and self-maintenance activities (10 items), fine motor activities (seven items), and gross motor playing activities (six items). It includes three categories 1) motor performance 2) daily participation and 3) learning of ADL. Motor performance is rated from 1 to 3 (1 = good performance, 2 = medium performance, 3 = poor performance), participation is rated from 1 to 4 (1 = the child does the activity regularly, 2 = the child does the activity sometimes, 3 = the child seldom or rarely does the activity, 4 = the child never does the activity); the learning subscale indicates the number of activities the child took longer to learn, ranging from 0 (the child did not take longer to learn any activity) to 23 (the child took longer to learn every activity). Norms are available based upon a Dutch and a Spanish sample. Higher scores indicate poorer performance, lower participation and longer learning. DCDDaily-Q is a valid and reliable questionnaire to address children's ADL performance [20, 30] and was used to answer research question 2.

**CSAPPA.** The CSAPPA was used to measure generalized self-efficacy. The 19-item questionnaire assesses three domains of self-efficacy; i) perceived adequacy (seven items); ii) predilection toward physical activity (nine items); and iii) enjoyment of physical education class (three items), has acceptable validity and test-retest reliability, and norms are based on a Canadian sample [31, 32]. The CSAPPA was used to answer research question 3.

## Statistical analysis

Statistical analyses were performed with SPSS 28.0. Demographic data (age, sex, height, weight, BMI) were used to describe the sample. ANOVA was used for comparison between the three groups on age, height, weight, and BMI; Chi-squared to compare the frequency of sex between groups.

Given large differences between groups in BMI, this variable was entered as covariate in the analysis. ANCOVA with post hoc pairwise comparison with Bonferroni correction, was performed to investigate differences in motor skill related fitness between the 3 groups. Power analysis showed that a sample of 90 TD and DCD children would be needed to show medium differences (d 0.5).

The questionnaire data (DCDDaily-Q, CSAPPA) showed a clear bimodal distribution and therefore we used non-parametric statistics. A non-parametric ANCOVA (QUADE) was used with BMI as covariate and with correction for multiple testing. Non-parametric correlations (Spearman rank order) were calculated between activities of daily living (DCDDaily-Q), self-efficacy (CSAPPA) and motor skill related fitness (PERF-FIT) for the TD and DCD groups, separately. Significance was set at $p < 0.05$.

## Results

### Participants

The descriptive statistics revealed that sex was equally distributed between TD and the two DCD groups (p = 0.275). No significant differences were found between groups in terms of age (p = 0.540) or height (p = 0.151). However, the differences in weight (F = 40.81 (2,213) p<0.001) and BMI were large (F = 71.43 (2,213) p<0.001), indicating that children with lower motor skills had higher BMI and that children with severe DCD were classified more frequently as overweight (Table 1 and Fig 2).

### Group differences regarding the motor skill-related fitness

Table 2 shows the statistics for the motor performance and agility and power items of the PERF-FIT for TD, m-DCD, s-DCD groups, and estimated mean with standard error (corrected for BMI). Children with DCD scored less proficient on all items of the PERF-FIT. Importantly, also the s-DCD scored significantly poorer than the m-DCD group. DCD groups also reached less far in the sit-and-reach test than TD.

### Group differences on the DCD-Daily-Q

Parents of children with DCD (both groups) reported that their child performed the mentioned activities less well (Fig 3).

DCD-Daily-Q total score showed a large group difference between TD and total DCD group (F = 97.55 (2,211) p = 0.001). Children with DCD participated less, and it also took them longer to learn the activities in comparison with their peers. This was the case for self-care and self-maintenance activities, fine motor activities, as well as gross motor playing activities. However, the differences between the two DCD groups were not significant. Performance ranged between 47–57 and 49–62 for the m-DCD and s-DCD group, respectively.

**Table 1. Shows the distribution of the demographic data among TD, m-DCD, s-DCD groups.**

|  | Age | | | Height (m) | | Weight (Kg) | | BMI | | Gender | |
|---|---|---|---|---|---|---|---|---|---|---|---|
|  | N | Mean | SD | Mean | SD | Mean | SD | Mean | SD | Male | Female |
| **TD** | 105 | 8.9 | 0.8 | 1.36 | 0.07 | 32.0 | 4.1 | 17.2 | 1.7 | 50 | 55 |
| **m-DCD** | 45 | 8.9 | 0.9 | 1.38 | 0.06 | 34.3 | 4.9 | 18.0 | 1.2 | 27 | 18 |
| **s-DCD** | 64 | 9.1 | 0.8 | 1.38 | 0.06 | 38.3 | 4.5 | 20.0 | 1.2 | 29 | 35 |
| **Total** | 214 | 9.0 | 0.8 | 1.37 | 0.06 | 34.4 | 5.2 | 18.2 | 1.9 | 106 | 108 |

N = Number of participants, SD = Standard Deviation, TD = Typically Developing, m-DCD = Moderate DCD, s-DCD = Severe DCD.

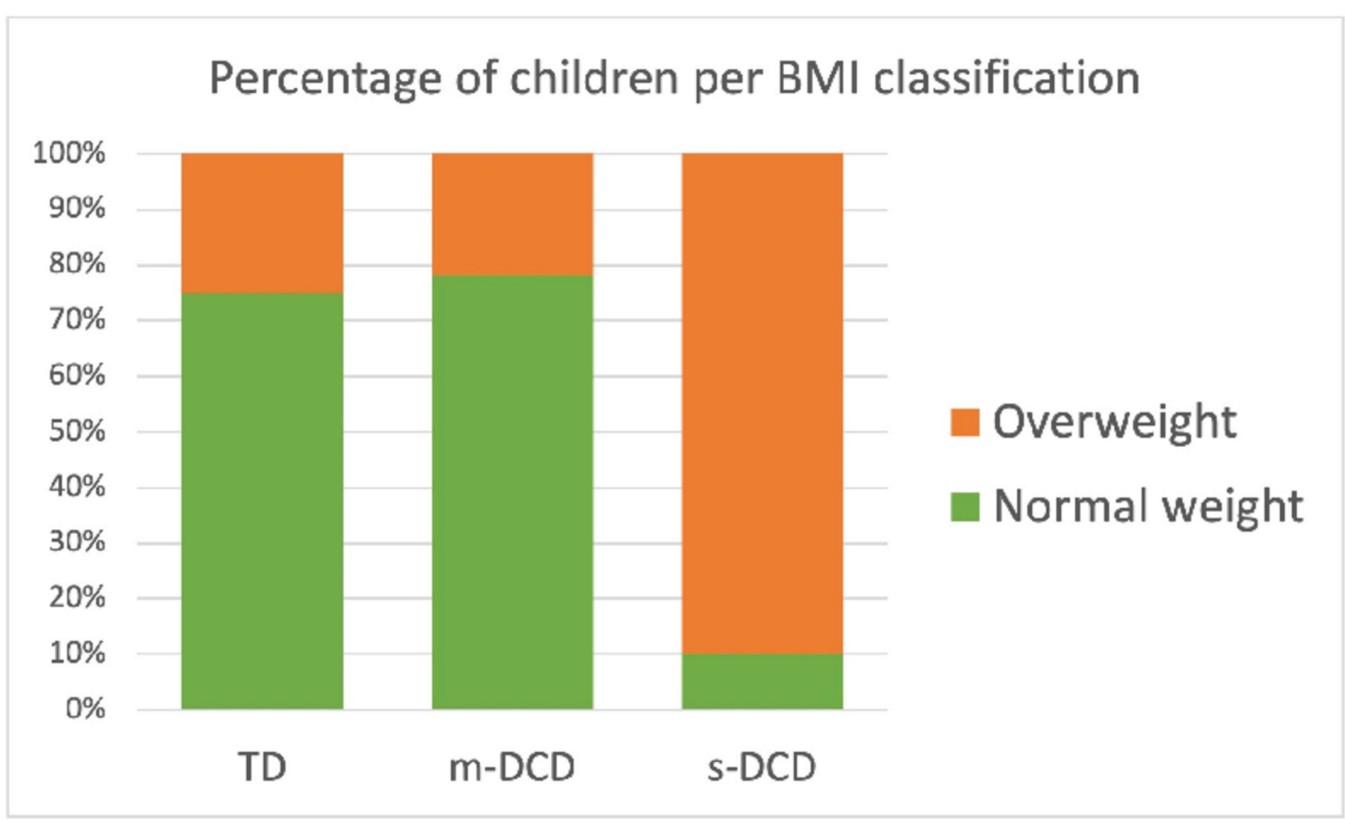

**Fig 2. Percentage per BMI classification for the 3 groups of children.** TD = Typically Developing, m-DCD = Moderate DCD, s-DCD = Severe DCD.

Participation scores range between 47–55 and 48–58 for the m-DCD and s-DCD group, respectively. The number of ADL that took a DCD child longer to learn ranged from 13–19 (m-DCD) and 13–20 (s-DCD) of the possible 23 items. Statistics are given in Table 3.

Table 4, gives details of the scores per domain of the DCDDaily-Q. Values are in accordance to the classification based on the norms [20]. Children belonging to m-DCD and s-DCD group all classified in the lowest 5 percent of "Quality of performance" and "Frequency of participation in ADL". Only the number of items that took children longer to learn is higher than 2 in some TD children, thus above the cut off for "no problems".

### Group differences on the CSAPPA

Large differences were found between the TD and DCD groups on all three domains according to answers of the children (adequacy in physical activity, enjoyment in physical education and predilection for physical activity). Total scores between groups were different (F = 95.75 (2,211) p = 0.001) (Fig 4). However, post hoc analysis showed no differences between m-DCD and s-DCD groups except for Adequacy. Statistics are given in Table 3. Table 4, gives details of the scores per domain of the CSAPPA.

### Association between performance of ADL (DCDDaily-Q), self-efficacy (CSAPPA) and motor skill-related fitness (PERF-FIT)

DCDDaily-Q and CSAPPA data showed an extreme bimodal distribution, which would lead to high correlations if calculated for the whole data set (Fig 5). Thus, only correlations within

**Table 2. Statistics for the comparison between groups on the PERF-FIT items, means and standard error are corrected for BMI.**

| PERF-FIT | Statistics | | TD | | m-DCD | | s-DCD | | Covariate BMI | Effect size of the group difference | Post hoc tests $ |
|---|---|---|---|---|---|---|---|---|---|---|---|
| Test items | F-value | p-value | Mean | SE | Mean | SE | Mean | SE | p-value | Eta | Pairwise comparison |
| **Power and Agility items** | | | | | | | | | | | |
| **Ladder Run (s)** | 88.92 | < .001 | 7.41 | .092 | 8.44 | .127 | 9.73 | .128 | .001 | .459 | all significant |
| **Ladder Step (s)** | 174.5 | < .001 | 14.31 | .103 | 16.29 | .142 | 17.81 | .142 | .001 | .624 | all significant |
| **Side Jump (#)** | 69.4 | < .001 | 20.40 | .254 | 18.16 | .35 | 14.76 | .352 | .002 | .397 | all significant |
| **Long Jump(cm)** | 79.59 | < .001 | 124.73 | 1.37 | 106.86 | 1.89 | 93.25 | 1.90 | .002 | .431 | all significant |
| **Overhead Throw (cm)** | 44.81 | < .001 | 238.18 | 1.89 | 231.89 | 2.60 | 205.34 | 2.62 | .038 | .299 | all significant |
| **Motor Performance items** | | | | | | | | | | | |
| **Bounce (#)** | 33.03 | < .001 | 41.96 | .251 | 40.64 | .346 | 38.09 | .348 | .076 | .239 | all significant |
| **Throw (cm)** | 64.99 | < .001 | 42.06 | .312 | 39.32 | .431 | 35.34 | .433 | .004 | .382 | all significant |
| **Jump/Hop (#)** | 143.96 | < .001 | 50.14 | .536 | 39.63 | .74 | 34.02 | .744 | .001 | .578 | all significant |
| **Static Balance (s)** | 65.76 | < .001 | 43.39 | .670 | 34.81 | .925 | 29.60 | .930 | .294 | .385 | all significant |
| **Dynamic Balance (#)** | 115.09 | < .001 | 22.87 | .370 | 15.48 | .511 | 13.54 | .514 | .291 | .523 | all significant |
| **Flexibility** | | | | | | | | | | | |
| **Sit and Reach (cm)** | 5.65 | .004 | 2.12 | .68 | .938 | .931 | min 2.17 | .936 | .939 | .051 | TD -DCD significant |

Df = Degrees of freedom, SE = Standard error, TD = Typically Developing, m-DCD = Moderate DCD, s-DCD = Severe DCD, eta = effect size, p-value<

0.05 = statistically significant, all significant = all pairwise comparisons are significant (1. TD vs m-DCD, 2. TD vs s-DCD, 3. m-DCD vs s-DCD), TD-DCD

significant = TD versus the combined DCD group comparison is significant. $ Post hoc tests with adjustment for multiple comparisons (Bonferroni).

the groups were examined. No significant correlations were found in the TD group between the answers of the parents in the DCDDaily-Q and the answers of the children in the CSAPPA. The time it took children to learn a skill according to their parents was significantly correlated in m-DCD with enjoyment and in s-DCD with the feeling of adequacy and predilection of the children (Table 5).

Because no differences were found in questionnaire scores between s-DCD and m-DCD correlations with PERF-FIT scores were calculated for the total DCD group and the TD group. Power and agility score of the PERF-FIT were related to participation in the TD group but no significant associations emerged between motor skill series level and self-efficacy or ADL results in children with good motor skills (Table 6).

In children with poor motor skills, participation, the time it took to learn a skill and predilection for physical activity were related to Power and Agility level. Additionally, participation, as reported by the parents, was related to motor skill series level in the DCD group. More ADL items that took longer to learn and lower participation coincided with poorer scores on Power and Agility. Data also suggests that the children with DCD may not be able to reflect on their own performance as they did not estimate their adequacy correctly. The correlation between perceived adequacy and actual performance on the motor skill series level was low and negative.

## Discussion

In this study, we intended to answer four research questions. Results showed that motor skill-related fitness is lower in the two groups of children with DCD than in TD and that the group at or below the 5th percentile scored worse than the group between the 9th and the 16th percentile.

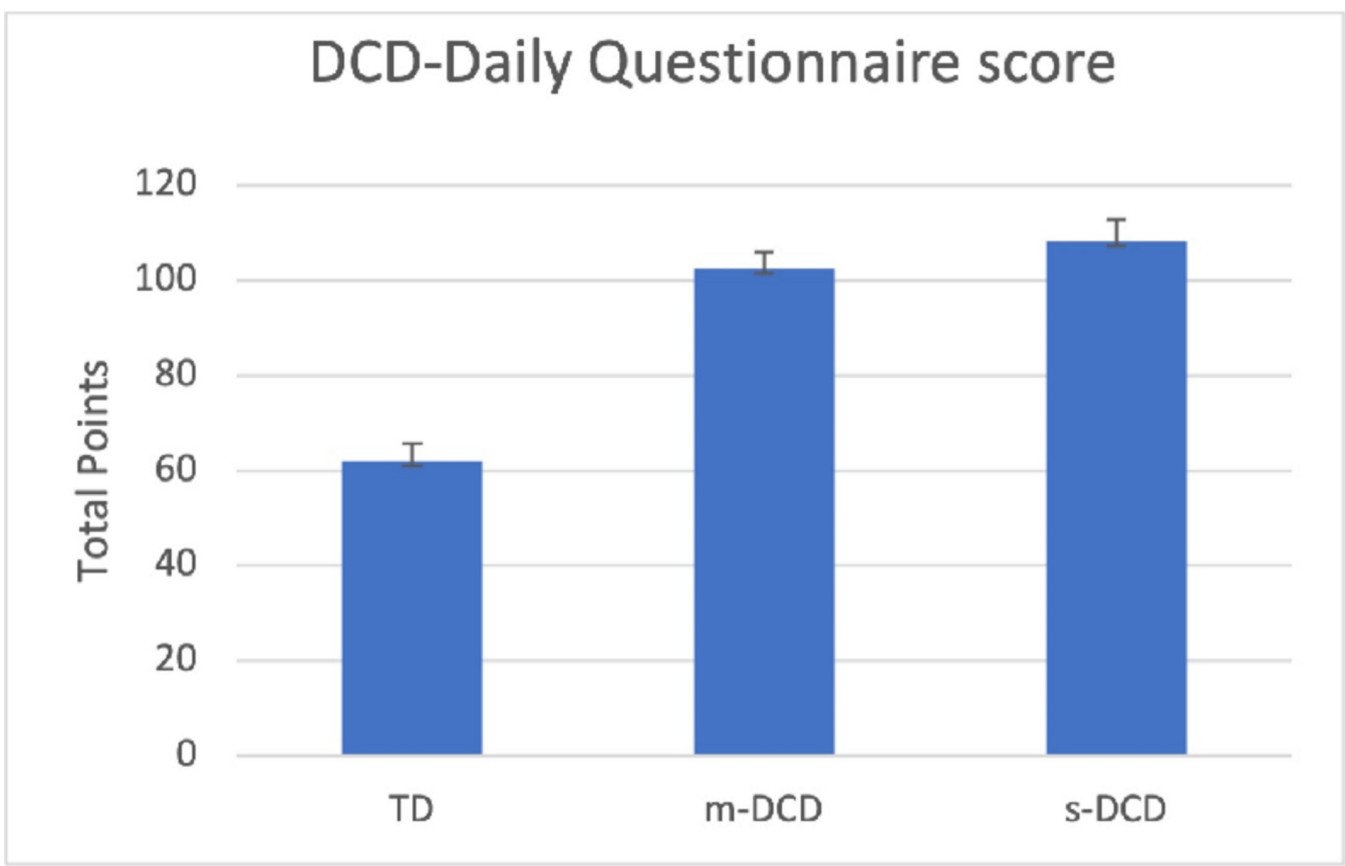

**Fig 3. The total score of the DCD-Daily-Q for the 3 groups of children.** Mean score (SD). TD = Typically Developing, m-DCD = Moderate DCD, s-DCD = Severe DCD. For statistics see Table 3.

**Table 3. Statistics (Non-parametric ANCOVA) for the DCDDaily-Q and CSAPPA total and domain scores.**

| Items | F-value | Main effect | TD vs m-DCD | TD vs s-DCD | m-DCD vs s-DCD |
|---|---|---|---|---|---|
| | | Groups p-value | p-value $ | p-value $ | p-value $ |
| DCD-Daily | | | | | |
| **DCD-Daily total** | 97.55 | 0.001 | 0.001 | 0.001 | 0.342 |
| **Performance** | 101.04 | 0.001 | 0.001 | 0.001 | 0.617 |
| **Participation** | 96.60 | 0.001 | 0.001 | 0.001 | 0.917 |
| **Learning** | 105.21 | 0.001 | 0.001 | 0.001 | 0.398 |
| CSAPPA | | | | | |
| **CSAPPA total** | 95.75 | 0.001 | 0.001 | 0.001 | 0.588 |
| **Adequacy** | 131.50 | 0.001 | 0.001 | 0.001 | 0.001 |
| **Predilection** | 100.18 | 0.001 | 0.001 | 0.001 | 0.255 |
| **Enjoyment** | 92.18 | 0.001 | 0.001 | 0.001 | 0.891 |

TD = Typically Developing, m-DCD = Moderate DCD, s-DCD = Severe DCD. $ Post hoc tests with adjustment for multiple comparisons.

**Table 4. Median and range of the scores for sections of the DCD-Daily-Questionnaire and domains of the CSAPPA per group.**

| | | DCDDaily-Q Performance | | | | Participation DCDDaily-Q | | | |
|---|---|---|---|---|---|---|---|---|---|
| Group | | Self care | Fine motor | Gross motor | Total | Self care | Fine motor | Gross motor | Total |
| TD n = 105 | Median | 14 | 8 | 7 | 30 | 14 | 8 | 10 | 3 |
| | Range | 13–16 | 7–12 | 6–9 | 27–35 | 10–16 | 7–12 | 7–13 | 0–5 |
| DCD n = 109 | Median | 25 | 16 | 13 | 53 | 25 | 13 | 14 | 16 |
| | Range | 20–28 | 13–19 | 11–17 | 47–61 | 19–28 | 11–18 | 11–16 | 13–20 |
| | | DCDDaily-Q Learning | | | | CSAPPA | | | |
| Group | | Self care | Fine motor | Gross motor | Total | | Adequacy | Predilection | Enjoyment |
| TD n = 105 | Median | 1 | 1 | 1 | 3 | | 20 | 28 | 10 |
| | Range | 0–2 | 0–2 | 0–3 | 0–5 | | 18–23 | 25–29 | 8–12 |
| DCD n = 109 | Median | 7 | 5 | 4 | 16 | | 16 | 20 | 7 |
| | Range | 5–9 | 3–7 | 3–6 | 13–20 | | 13–18 | 16–24 | 4–9 |

Note that there is no overlap in the scores between TD and DCD groups for the DCDDaily-Q and for predilection and enjoyment of the CSAPPA. Lower values indicate better scores for DCDDaily-Q and for the CSAPPA Higher values indicate better scores. N = Number of participants, TD = Typically Developing, DCD = Developmental Coordination Disorder.

Children with DCD, irrespective of the severity, have robust difficulties with motor learning, and motor performance, which significantly interfere with their activities of daily living. Next, we have shown that parents rated activities of daily living much lower in children with

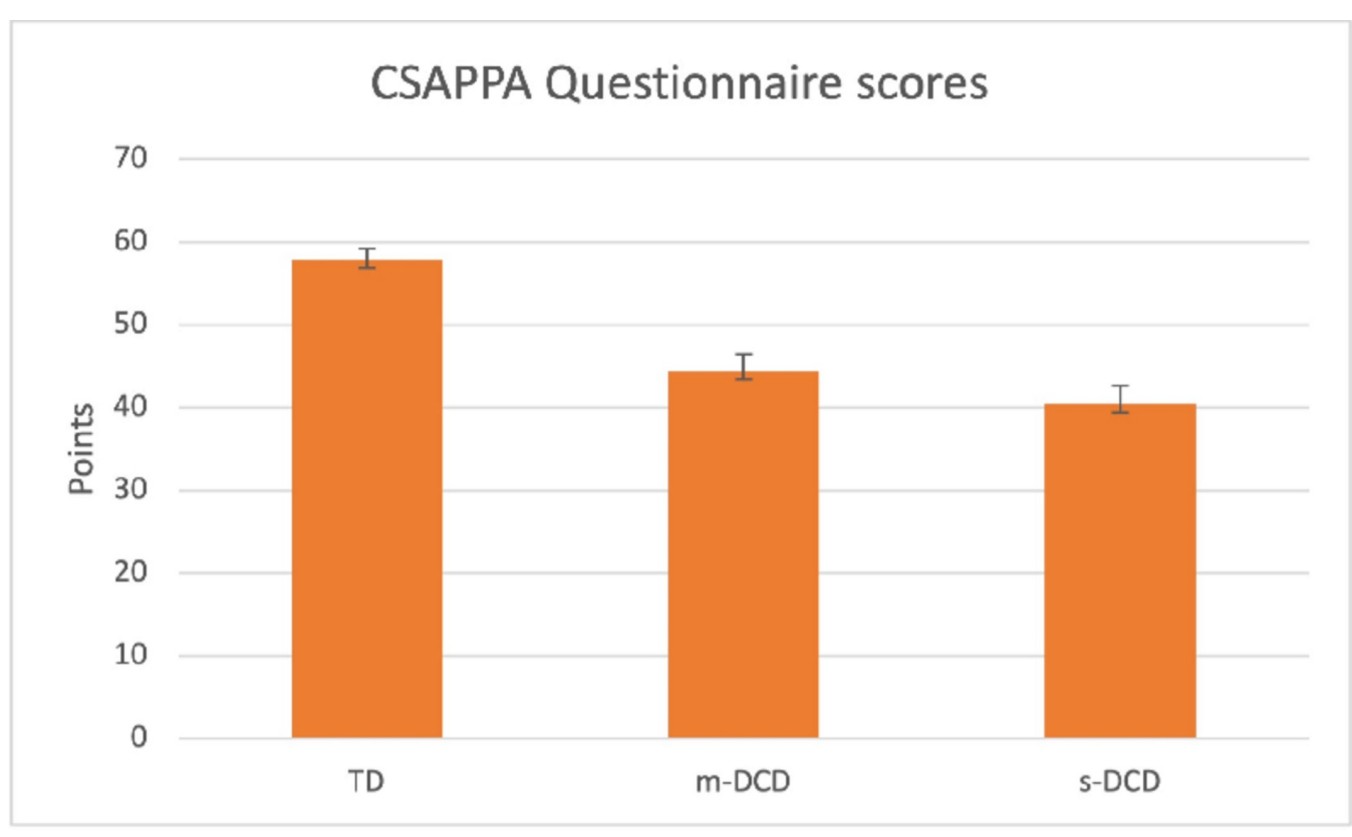

**Fig 4. The total score of the CSAPPA for the 3 groups of children.** Mean score (SD). TD = Typically Developing, m-DCD = Moderate DCD, s-DCD = Severe DCD. For statistics see Table 3.

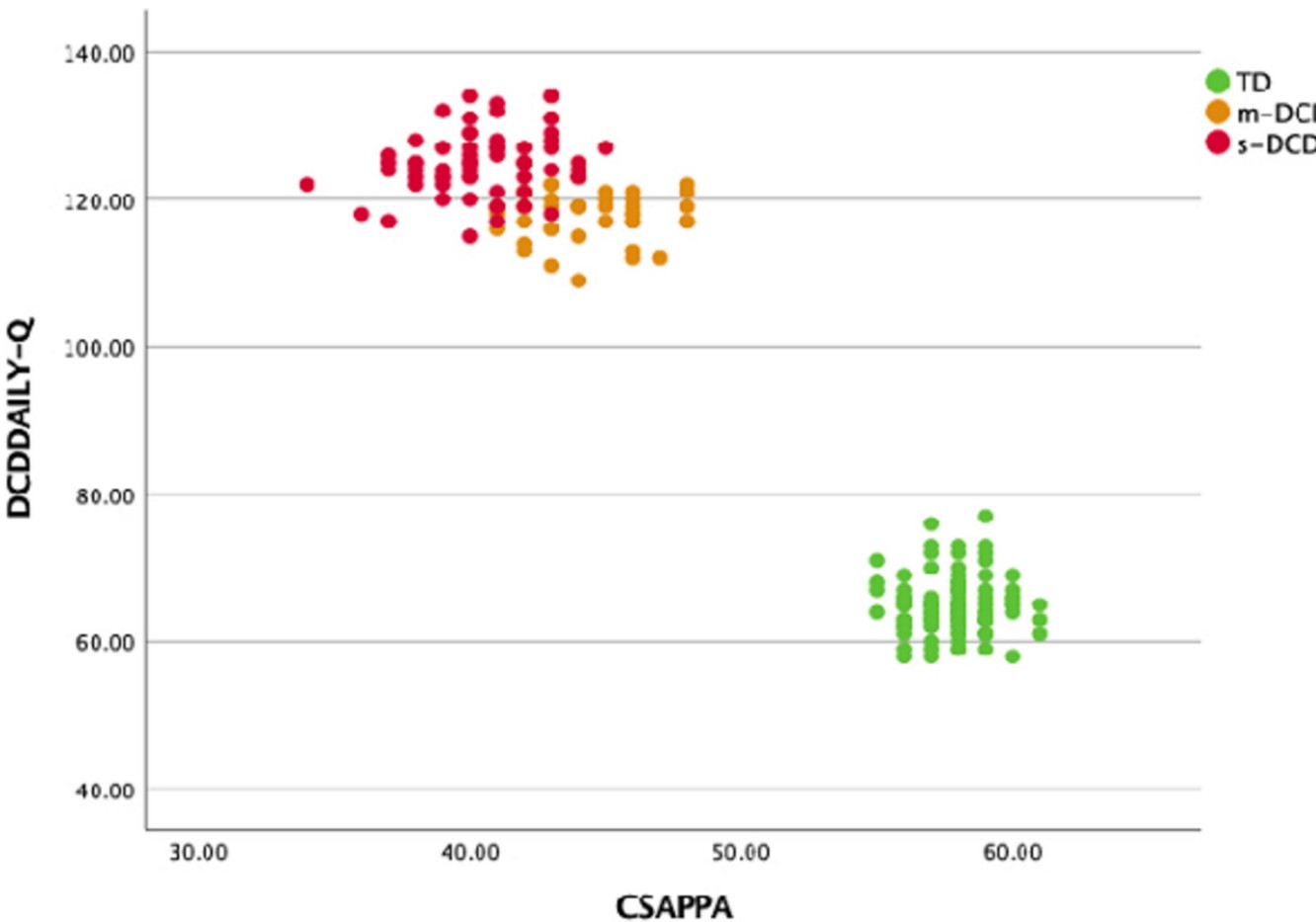

**Fig 5. Scatterplot with total score of DCDDaily-Q on the y-axis and for CSAPPA on the x-axis.** Note that there is no overlap in scores for the TD children and the two groups of children with DCD. TD = Typically Developing, m-DCD = Moderate DCD, s-DCD = Severe DCD.

DCD, but level of motor impairment was not a significant factor. The same result was found for self-perception; it didn't matter for the enjoyment or predilection for physical activity to which DCD group children belonged, they scored very poorly. Finally, the association between

**Table 5. Non-parametric correlations between domain scores of DCDDaily-Q and CSAPPA per group.**

| Group | Items | CSAPPA Adequacy ($r_s$) | CSAPPA Predilection ($r_s$) | CSAPPA Enjoyment ($r_s$) |
|---|---|---|---|---|
| TD (n = 105) | DCDDaily-Q performance | .064 | -.115 | -.080 |
| | DCDDaily-Q participation | .060 | -.013 | -.020 |
| | DCDDaily-Q learning | .042 | .040 | -.147 |
| m-DCD (n = 45) | DCDDaily-Q performance | .201 | -.064 | .089 |
| | DCDDaily-Q participation | -.008 | -.116 | .010 |
| | DCDDaily-Q learning | .210 | -.281 | -.311* |
| s-DCD (n = 64) | DCDDaily-Q performance | .085 | .027 | -.095 |
| | DCDDaily-Q participation | .241 | .113 | .218 |
| | DCDDaily-Q learning | .322** | .267* | .007 |

TD = Typically Developing, m-DCD = Moderate DCD, s-DCD = Severe DCD, $r_s$ = Spearman's rho, p-value< 0.05 = statistically significant * p<0.05, ** p<0.01.

**Table 6. Nonparametric correlations of DCDDAILY-Q and CSAPPA with PERF-FIT scores.** Note correlations are expected to be negative between DCDDAILY-Q and PERF-FIT, and positive for CSAPPA and PERF-FIT.

| Group | | Power and | Motor | PERF-FIT |
|---|---|---|---|---|
| | | Agility ($r_s$) | skill series ($r_s$) | Total ($r_s$) |
| **TD n = 105** | DCDDaily-Q performance | -.037 | -.040 | -.049 |
| | DCDDaily-Q participation | -.242* | -.130 | -.218* |
| | DCDDaily-Q learning | .110 | .132 | .148 |
| | DCDDaily_Q Total | -.197* | -.066 | -.151 |
| | CSAPPA Adequacy | -.071 | .123 | .072 |
| | CSAPPA Predilection | .158 | .049 | .108 |
| | CSAPPA Enjoyment | .064 | .046 | .057 |
| | CSAPPA Total | .100 | .153 | .163 |
| **DCD n = 109** | DCDDaily-Q performance | -.161 | -.065 | -.179 |
| | DCDDaily-Q participation | -.447** | -.252** | -.468** |
| | DCDDaily-Q learning | -.226* | -.133 | -.239* |
| | DCDDaily_Q Total | -.382** | -.234* | -.416** |
| | CSAPPA Adequacy | .070 | -.226* | -.076 |
| | CSAPPA Predilection | .275** | .108 | .266** |
| | CSAPPA Enjoyment | .127 | .117 | .152 |
| | CSAPPA Total | .241* | .028 | .198* |

n = Number of participants, TD = Typically Developing, DCD = Moderate and Severe DCD, $r_s$ = Spearman's rho, * $p<0.05$, ** $p<0.01$.

information obtained via the questionnaires filled out by parents (DCDDaily-Q) and by the children (CSAPPA) was low within the TD and DCD group. The highest relation was found between the speed of motor learning indicated by the parents and enjoyment in physical education rated by the children.

## Motor skill-related fitness

The results indicated lower levels of motor skill-related fitness in children with DCD, especially children below the 5th percentile both on the Power and Agility component as well as the Motor Performance component (Ball skills, Balance, Hopping). In recent decades, extensive research has revealed deficiencies in various fundamental motor skills [2] such as balance [33] and ball skills [34, 35]. The Power and Agility results are also similar to findings in earlier studies that indicated compromised explosive power, muscle strength [36, 37] and endurance [36, 38].

Flexibility was assessed using the sit-and-reach test, which measures the flexibility of hamstrings and lower back, and not overall joint flexibility [29]. We found lower flexibility in the children with DCD, while Jelsma et al. [39] reported higher prevalence of hypermobility in the DCD-group. Most likely, this is caused by the different outcomes used; Beighton score versus Sit and Reach. To get valid information about the level of hyper- or hypomobility, it is recommended to measure range of motion of the most important joints [40].

## BMI

Our results also showed that children with DCD had higher BMI than TD children. This corresponds to various studies that found similar results [13, 36]. Farhat et al. [36] reported a significant relationship between physical ability and BMI in children with DCD. Cairney et al. [13] reported higher body fat percentages in children with DCD than in TD children (23.3% vs.12.1%). High BMI is also associated with poor explosive power and exercise tolerance in

children with DCD, and this relation was not observed in TD children [36]. Based on the inter-dependence, exercise tolerance, BMI and explosive power are significant predictors of motor coordination and fitness levels [36]. Fitness and obesity have already been reported as major concerns for DCD in the USA and Israel [41], but it seems that African children in our study with high or low motor competence were also different from each other in fitness levels and BMI. Importantly, there is already an increase in the prevalence of metabolic syndrome on the African continent due to departure from traditional African to western lifestyles [42]. Non-communicable diseases, as a result of inactivity, will put an extra economic burden on the health care systems in low- and middle-income countries in the near future [43].

## Participation

Literature has widely reported that poor motor skills reduce participation in children with DCD [19, 44]. Our data suggest that children's motivation to participate in leisure sports is significantly related to their level of skill in motor performance, which was also reported by Larsen et al. [42]. This is most likely due to their lower level of motor skill.

The relation between self-efficacy and DCDDaily-Q was different in the two DCD groups. The highest association with motor learning as reported by the parents was found with perceived enjoyment of physical education by the child in the moderate DCD group, if it took them longer to learn motor skills, they would like PE classes less. In the severe DCD group, the highest association with the time to learn skills emerged with perceived adequacy in physical education. The longer it took them to learn motor skills, the lower their perceived adequacy in physical education was. In children with good motor skills, however, motor skill level was not related to self-efficacy or ADL results, while fitness levels were. In fact, in both groups the highest correlation between predilection to physical activity and participation was seen for the power and agility scale of the PERF-FIT and not for the motor skill item series. This might be explained by findings in an earlier study that revealed the scores on the PERF-FIT are significantly related to performance in active play [6]. Outdoor play and sport-like games in children are characterized by short periods of intensive physical activity, interspersed with short periods of reduced or less intensive activity [45]. If children have lower levels of explosive muscle power, anaerobic capacity and agility, this will lead to restrictions in participation [38, 46]. Yet in intervention, we predominantly focus on improving motor skills and less on preparing the children for adequate levels of sports participation [47]. Additionally, the results indicated that children with DCD scored poorly on self-efficacy regardless to the severity of the motor deficit. This supports other studies that found that children with low motor proficiency usually avoid engaging in motor activities not only because they don't have the skills but also because they dislike doing it [14, 48]. Consequently, the low adequacy and predilection to physical activity will eventually widen the motor skill and the health status gap between these children and their peers, and increase the risks for more sedentary behavior. This will further raise the prevalence of children being overweight or obese and their negative perceptions of physical activity which puts them at risk for developing pathologies adversely affecting cardiovascular health, pulmonary health, endocrine function, and mental health [47].

## Test outcomes and questionnaire information

Questionnaires may be subjective, but as shown, they provide a valuable source of information of the perceived problems by parents (DCDDaily-Q) and children (CSAPPA) in our study. Importantly, they give the clinician different information, confirmed by the low association between parental and child responses within the DCD and TD group. Results confirm that the children with DCD had a difficult time learning most of the motor skills during development,

which information could be used to confirm criterion C. Moreover, the questionnaire data show that poor motor skill may exclude the children from participating in important activities of daily living, a fact which concurs with earlier studies [49].

Except for psychometric evaluation, not many studies have used the DCDDaily-Q to address children's performance in ADL [20]. Van der Linde et al. [20] found that delays in motor learning predicted poor motor performance in children with DCD, which in turn predicted less daily participation in both groups. Our research suggests that the DCDDaily-Q has an extraordinary discriminant capacity. When the provided cut off values were used, all children were classified right as TD or DCD and hence the tool can effectively identify children at risk of DCD [50] (see Fig 5).

## Impact of poor motor skill related fitness

Good physical fitness is not only important for physical health but also for mental health and social-emotional well-being. As a consequence of their poor motor skills, children with DCD often avoid physical activities, which means that they not only end up in social isolation but also develop lower self-esteem, self-concept, and self-efficacy in comparison to children without DCD. Children with low motor competence perceived themselves to be less competent in basic motor skills as well as in their overall physical abilities [14]. However, children's perception is far from perfect, which was shown by the lacking relation between CSAPPA Adequacy and actual measured motor skills and fitness. Remarkably, although low, the association of CSAPPA Predilection and total CSAPPA with PERF-FIT subscale was higher for Power and agility than for the subscale Performance including catching, hoping and balance items. This may be because predilection to physical activity is related to leisure sports, which require fitness. Moreover, for the DCDDaily-Q participation the highest correlation was also found with Power and agility. Indeed, children who participated more in physical activities had a better ability to rapidly change direction. Noordstar and Volman [51], used the Self-Perceptions Profile for children and reported that children with DCD had significantly lower perceptions of their athletic competence and perceive themselves lower in social acceptance when compared with TD children [51]. Children with DCD may be the last to be asked to join games or sport or may not even participate at all [51]. Hence, children with DCD spend more time as spectators and are often excluded from leisure activities due to their low motor competence [52]. Our results are supported by Cairney et al., [16] who reported that children with DCD had lower participation in free play and organized activities than TD children. Children with DCD are at a higher risk for internalizing problems, including withdrawal from social interactions, being emotionally reactive and have more somatic complaints [17, 53]. Encouragingly, Li et al. [54] reported that physical activity has a positive effect on internalizing problems in children with DCD. Our data show that children at a young age (7–10), already have poor perceived physical competence, reduced motivation to participate in physical activity, and consequently fewer opportunities to develop proficient motor skills and adequate fitness levels, which was already shown to persist in older children [55]. Thus, these factors should be considered when developing interventions.

## Practical implications

This study showed that components of motor skill-related fitness are not attainable for individuals with low motor competence. Hence, adapted training to prevent later chronic disease is indicated in which training principles should be followed but starting with low coordination requirements and still adequate levels of challenge. Exercise enjoyment differed between children in the low and high motor competence group, which is a threat for their future

engagement. Children need to start being physically active at a young age and have positive experiences to develop lifelong adherence to exercise [56]. It is important for children to gain proficiency in fundamental motor skills as early as possible since the relationship between motor ability and participation in physical activity may strengthen over time [57]. In line with the finding of Smits-Engelsman et al. [58] in a group of Brazilian children, the present study confirmed how important it is for clinicians to follow up on skill-related fitness to address potential future adverse health effects as the levels of performance and participation seen in children with DCD may not be sufficient to develop and maintain a healthy lifestyle. Thus, improving motor skill-related fitness in children with DCD is urgently needed. The overall health benefits associated with the development and maintenance of motor skill-related fitness should be taken into account when prescribing intervention for children with DCD [19].

Exercise is medicine, especially in children with low adequacy and predilection for physical activity. There are numerous playful functional activities that can affect the musculoskeletal system and can be used in children with low motor skills such as hiking, jogging, active transportation, play structure climbing, swimming, dance and aerobics with simple coordination patterns, to name a few. Also, sport with adapted rules and number of players should be tried out and as well as structured use of active videogames [59]. Gaining better motor skills and fitness may contribute to children's enjoyment and participation in physical activity and vice versa. Because physical activity also has social participatory elements such as playing with peers during school recess, it provides children with opportunities to practice social skills including communication skills, taking turns, and conflict resolution [50].

## Limitation and future studies

The major limitation of this study is that hardly any tool or questionnaire is validated for African populations. Thus, cultural bias may have influenced the results. Although the DCDDaily-Q norms for Spain did not diverge much from Dutch norms [50], this needs further study for other cultures. Also, the MABC-2 is not validated or normed for African children. Since the study was conducted among children living in low-income areas, perhaps their low socioeconomic status gave them less opportunity to develop the skills required for optimal performance on the MABC-2 test items [60]. So, the results might be different in groups living in more affluent areas that have more practice opportunities.

Moreover, concerning criteria D, intellectual or cognitive impairment, was evaluated by interviews with parents or teachers and not measured with an IQ-test. Neither were children seen by a neurologist to rule out other medical conditions that could explain impaired motor development. Thus, it might be possible that some children had unnoticed cognitive impairment or comorbidities.

There are some gaps that can be researched in future studies. Longitudinal studies are needed to examine the effect of improved motor skills and fitness after intervention on perceived adequacy and participation. More research is also needed on joint mobility in children with DCD, using valid tools. There are currently conflicting narratives on whether children with DCD have more or less flexibility than their peers and importantly whether this influences posture, motor skills, and fitness.

## Conclusion

Children with DCD are less physically fit and do not like physical activity. According to parents, they needed more time than peers to learn the ADL items mentioned in the DCDDaily-Q. Their motor skill-related fitness is compromised, which can significantly affect physical and mental health. Understanding parent's and children's perceptions about daily activities

seems to be an important adjunct to clinical management. Low motor skill-related fitness in children with DCD and the fact that they do not enjoy physical activity is an important concern for their present and future health status, of which clinicians should be aware.

## Supporting information

**S1 Data.**
(SAV)

## Acknowledgments

We acknowledge the support of parents, children, and management of the participating schools. The researchers would like to acknowledge Deanship of Scientific Research, Taif University, Saudi Arabia for supporting this study.

## Author Contributions

**Conceptualization:** Faiçal Farhat, Adnene Gharbi, Haithem Rebai, Wassim Moalla, Bouwien Smits-Engelsman.

**Data curation:** Faiçal Farhat, Mohamed Moncef Kammoun, Adnene Gharbi, Haithem Rebai, Wassim Moalla, Bouwien Smits-Engelsman.

**Formal analysis:** Faiçal Farhat, Marisja Denysschen, Nourhen Mezghani, Mohamed Moncef Kammoun, Haithem Rebai, Bouwien Smits-Engelsman.

**Investigation:** Faiçal Farhat, Marisja Denysschen, Wassim Moalla, Bouwien Smits-Engelsman.

**Methodology:** Faiçal Farhat, Nourhen Mezghani, Mohamed Moncef Kammoun, Adnene Gharbi, Haithem Rebai, Bouwien Smits-Engelsman.

**Project administration:** Faiçal Farhat.

**Resources:** Faiçal Farhat.

**Software:** Faiçal Farhat.

**Supervision:** Faiçal Farhat.

**Writing – original draft:** Faiçal Farhat, Marisja Denysschen, Nourhen Mezghani, Bouwien Smits-Engelsman.

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
