## [Decision Letter · Decision Letter 0]

16 Oct 2023

PONE-D-23-30683Activities of daily living, self-efficacy and musculoskeletal fitness are compromised in children with developmental coordination disorderPLOS ONE

Dear Dr. Farhat,

Thank you for submitting your manuscript to PLOS ONE. After careful consideration, we feel that it has merit but does not fully meet PLOS ONE’s publication criteria as it currently stands. Therefore, we invite you to submit a revised version of the manuscript that addresses the points raised during the review process.

We look forward to receiving your revised manuscript.

Kind regards,

Mohamed Rafik N. Qureshi, Ph.D.

Academic Editor

PLOS ONE

Journal Requirements:

Additional Editor Comments:

To comply with PLSO ONE's quality standards, the manuscript entitled "Activities of daily living, self-efficacy, and musculoskeletal fitness are compromised in children with developmental coordination disorder" needs to be modified. The necessary data file may be uploaded along with the questionnaires used.

Reviewers' comments:

Reviewer's Responses to Questions

**Comments to the Author**

1. Is the manuscript technically sound, and do the data support the conclusions?

Reviewer #1: Partly

Reviewer #2: Yes

2. Has the statistical analysis been performed appropriately and rigorously? 

Reviewer #1: Yes

Reviewer #2: Yes

3. Have the authors made all data underlying the findings in their manuscript fully available?

Reviewer #1: Yes

Reviewer #2: No

4. Is the manuscript presented in an intelligible fashion and written in standard English?

Reviewer #1: Yes

Reviewer #2: Yes

5. Review Comments to the Author

Reviewer #1: The aim of the study is to determine musculoskeletal fitness level, the level of activities of daily living in children (ADL) as reported by their parent and self-efficacy as reported by the children with moderate and severe DCD and compare that to typically developing children (TD). Overall, the paper reads very well, is well conducted, and includes a large group of children with DCD meeting all diagnostic criteria. That physical fitness, ADL and self-efficacy are compromised in DCD has been reported in previous studies, although not in an African country, and not related to different levels of motor impairment. However, the relation between the three concepts has hardly been research before. I would recommend to focus more on this relation in the introduction. At present, the introduction mainly focusses on compromised physical fitness and poor-self efficacy. Research question 2 needs more justification than presently given, as the need to study ADL is hardly mentioned in the introduction. The same holds for research question 4: why is it important to study the relationship between these concepts? And what kind of relationship do you expect based on theory/previous research? The impact of the paper would increase if those 2 research questions are more thoroughly introduced.

Title/abstract: I would suggest to include the relationship between the different concepts in the title and abstract.

Introduction: see above: rationale for research questions 2 and 4 is lacking.

- Line 92: ‘needed sports’ � ‘needed for sports’

Methods:

- Line 122-123: The DCDDailyQ and CSAPPA were translated into the Arabic language. Was the backward-forward translation method applied?

- Line 126: was the CSAPPA individually administered to the children? Please explain in the text.

- Line 141: ‘parental questonnaire’. This questionnaire has not been mentioned before. What kind of questionnaire was used?

- Line 147: how many children were selected by teachers, and how many met the diagnostic criteria for DCD?

- DCDDaily and CSAPPA: please add that only Dutch/Canadian norms were available for these instruments.

- Line 198-199: “Non-parametric correlations were calculated for the TD and DCD groups, separately”. Please add between which variables correlations were calculated

- Considering the number of post-hoc tests: was a Bonferoni correction applied?

Results:

- Line 205: “The descriptive statistics revealed that sex was equally distributed between TD and DCD groups (p=0.275)”. Did you also check whether sex was equally distributed between the two DCD groups? At first glance it seems that far more girls than boys were in the sDCD group compared to the mDCD group.

- Table 1: Height: please present two figures after the full stop.

- Table 2: please add measurement units for the PERFIT subtests. Please add in column 3 what the p-values represent (difference between TD and DCD groups?). I was confused by the label BMI, please explain.

- The DCDDailyQ is written in different ways throughout the text (DCD-DAILY-Q; DCDDaily-Q). In the papers by Van der Linde, the author of the questionnaire, it is called the DCDDaily-Q.

- Line 299: “Data also suggest that the CwDCD did not estimate their adequacy correct” This is not what the data reveal. Children with CwDCD rated their adequacy comparable to the TD group. This does not have to mean that it is incorrect. Young children may not be able to reflect on their own performance. The same statement also occurs in the discussion. Please reflect on this.

Discussion:

- Lines 305-317: I miss a summary of the relation between the PERF-FIT and the questionnaires, where some interesting relations were reveled for the DCD group between Participation/predilection and power/agility and motor skills.

- Lines 318 and following: I miss a reference to the review of Rivilis (Rivilis I, Hay J, Cairney J, Klentrou P, Liu J, Faught BE. Physical activity and fitness in children with developmental coordination disorder: a systematic review. Res Dev Disabil. 2011 May-Jun;32(3):894-910. doi: 10.1016/j.ridd.2011.01.017. PMID: 21310588.)

- Line 323: “Flexibility was assessed using the sit-and-reach test, which measures the flexibility of hamstrings and lower back, and not overall joint flexibility [26]”. This sentence should be added to the methods section, where information about the sit-andreach is scarce.

- Line 325: “We recommend using goniometry of the most important joints, to get valid information about the level of hyper- or hypomobility [27]”. This comment comes out of the blue, as hyper- or hypomobility is not discussed before.

- Line 327: What I miss in the discussion about BMI is the fact that the BMI levels of the children in the present study are still within the normal range. The text know reads as if the children with DCD are overweight, which is not the case.

- Line 342: “Our data suggest that children’s motivation to participate in leisure sports is significantly influenced by their level of skill in motor performance”. This is too bold, as no causal directions were tested. Children with DCD indicated to be less motivated to participate in leisure sports. This may be due to their lower level of motor skill.

- Line 345-347: “In fact, in both groups the highest correlation between predilection to physical activity and participation was seen for the fitness scale of the PERF-FIT and not for the motor skill item series.” Do you have an explanation? Maybe because predilection to physical activity is related to leisure sports, which require fitness?

- I miss a discussion about the differences between TD and DCD groups regarding ADL and CSAPPA data, and how they relate to previous studies. You refer to Noordstar and Volman, but not in relation to your own findings.

- Line 396: “This study showed that components of health-related fitness are not attainable for individuals with low motor competence”. I disagree. Your study shows that children with DCD have lower fitness levels, not that health-related fitness levels are not attainable. I do hope that these children can improve with training.

- I miss in the discussion a remark about the different boys-girls ratios in the mDCD and sDCD groups and how this may (or not) have affected your results.

Reviewer #2: Title of the paper: Activities of daily living, self-efficacy and musculoskeletal fitness are compromised in children with developmental coordination disorder

Developmental co-ordination disorder (DCD) is leading to reduced physical co-ordination and increased risk towards health problems. The paper determines the musculoskeletal fitness level in children with moderate and severe DCD and compare that to typically developing children (TD). It also finds the level of activities of daily living (ADL) as reported by their parent and self-efficacy as reported by the children. The study compares TD children (n=105) and children with DCD (n=109; 45 moderate DCD and 64 severe DCD) of Tunisian students aged between 7 and 10 years of age. The DCD Daily-Questionnaire and Children’s Self-perceptions of Adequacy in and Predilection for Physical Activity Questionnaire were used to determine ADL and adequacy towards physical activity, respectively. The PERF-FIT was used to measure musculoskeletal fitness levels.

Comments:

1) “Activities of daily living, self-efficacy and musculoskeletal fitness are compromised in children with developmental coordination disorder” may be changed to “Activities of daily living, self-efficacy and musculoskeletal comparison among the children with developmental coordination disorder”

2) Please refer to:” In total 214 children participated in the study: 105 TDC and 109 CwDCD. How the sample size was determined is unclear. Sample size selection may be verified using Lwanga and Lemeshow (1990).

Lwanga, S.K., Lemeshow, S. and World Health Organization, 1991. Sample size determination in health studies: a practical manual. World Health Organization.

3) Author may provide flow diagram research methodology steps for sample recruitment and follow up for more clarity.

4) The legend shown below Table 1 may be changed, M-DCD may be changed to m-DCD, similarly S-DCD may be changed to s-DCD to make uniform in throughout the manuscript.

5) Table 2 may be properly formatted to identify the heading of Power and Agility, Motor item series, and Flexibility

6) In Table 2 The heading F may be F-value, legend and last column entry "all sign" may be "all significant" since "all sign" has misleading meaning, similarly "TD-DCD sign" may be "TD-DCD significant"

7) Table 3 first column suitable heading may be provided for instance. Test description. The dF2,211 may be added at the table bottom. The degree of freedom (DF) may be used throughout the manuscript.

8) Table 4 first two columns may be suitably named as DCD types and Statical parameters respectively.

9) Notes below the Tables may be suitable changed to small font size or maybe in italics. All notes must be below adjoining the Table.

10) Some typos may be checked for instance "table 3" should be 'Table 3", "Table 4. gives..." should be “Table 4, gives..." "...and predilection of the children. (Table 5)." should be "...and predilection of the children (Please refer Table 5)." "by Larsen et al. [29]" should be "by Larsen et al. [29]."

11) Table header for Table 5 may be suitable changed. The values of Spearman's rho are shown in the last three columns, The first two columns must be suitable named.

12) Table header for first two columns of Table 6 may be suitable added.

13) Asper Jelsma et al. (2013), ‘In general the motor performance and joint mobility are not related, in DCD.” Hence recommending the use of goniometry may be disjoint here.

Jelsma, L.D., Geuze, R.H., Klerks, M.H., Niemeijer, A.S. and Smits-Engelsman, B.C., 2013. The relationship between joint mobility and motor performance in children with and without the diagnosis of developmental coordination disorder. BMC pediatrics, 13, pp.1-8.

14) Some recommendations put forward by Mandich et al., (2001) as " Skill acquisition through evidence-based practices, Interventions leading to functional outcomes etc. may also be discussed.

Mandich, A.D., Polatajko, H.J., Macnab, J.J. and Miller, L.T., 2001. Treatment of children with developmental coordination disorder: What is the evidence?. Physical & occupational therapy in pediatrics, 20(2-3), pp.51-68.

15) Some similar studies related to PERF-FIT, DCD may be compared with the present studies:

• Smits-Engelsman, B., Neto, J.L.C., Draghi, T.T.G., Rohr, L.A. and Jelsma, D., 2020. Construct validity of the PERF-FIT, a test of motor skill-related fitness for children in low resource areas. Research in developmental disabilities, 102, p.103663.

• Girish S, Raja K, Kamath A. Prevalence of developmental coordination disorder among mainstream school children in India. Journal of pediatric rehabilitation medicine. 2016 Jan 1;9(2):107-16.

• Montes-Montes, R., Delgado-Lobete, L., Pereira, J., Schoemaker, M.M., Santos-del-Riego, S. and Pousada, T., 2020. Identifying children with developmental coordination disorder via parental questionnaires. Spanish reference norms for the DCDDaily-Q-ES and correlation with the DCDQ-ES. International Journal of Environmental Research and Public Health, 17(2), p.555.

• van der Linde, B.W., van Netten, J.J., Otten, B., Postema, K., Geuze, R.H. and Schoemaker, M.M., 2013. Development and psychometric properties of the DCDDaily: a new test for clinical assessment of capacity in activities of daily living in children with developmental coordination disorder. Clinical Rehabilitation, 27(9), pp.834-844.

Data availability:

16) As authors all relevant data are within the manuscript and its Supporting Information files. However. no data file is uploaded. Authors also need to upload DCD Daily-Questionnaire and Children’s Self-perceptions of Adequacy in and Predilection for Physical Activity Questionnaire.

6. PLOS authors have the option to publish the peer review history of their article (what does this mean?). If published, this will include your full peer review and any attached files.

Reviewer #1: No

Reviewer #2: No

---

## [Author Response · Author response to Decision Letter 0]

8 Nov 2023

Reviewer 1 

1. The aim of the study is to determine musculoskeletal fitness level, the level of activities of daily living in children (ADL) as reported by their parent and self-efficacy as reported by the children with moderate and sever DCD and compare that to typically developing children (TD). Overall, the paper reads very well, is well conducted, and includes a large group of children with DCD meeting all diagnostic criteria. That physical fitness, ADL, and self-efficacy are compromised in DCD has been reported in previous studies, although not in an African country, and not related to different levels of motor impairment. However, the relation between the three concepts has hardly been researched before. I would recommend to focus more on this relation in the introduction. At present, the introduction mainly focuses on compromised physical fitness and poor self-efficacy. Research question 2 needs more justification than presently given, as the need to study ADL is hardly mentioned in the introduction. The same holds for research question 4: why is it important to study the relationship between these concepts? And what kind of relationship do you expect based on theory/previous research? The impact of the paper would increase if those 2 research questions are more thoroughly introduced. 

Response 

We thank the reviewer for her valuable and detailed comments which helped to improve the paper. New information about the relationship between these concepts has been added to the abstract, intro and discussion. So, for details please see below.

2. Title/abstract: I would suggest to include the relationship between the different concepts in the title and abstract 

Response 

We have adapted the title to accommodate for this comment. 

3. Added info to the Abstract to a max of the 300 words. 

Response 

Small adaptation have also been made in the introduction. 

4. Introduction: See above: rationale for research questions 2 and 4 is lacking 

Response 

We think the rationale for the study has been described at several points in the introduction. For instance the text reads” “However, little is known, about the children’s difficulties in specific activities of daily living (ADL) and participation in ADL [17]. Moreover, the scares evidence available comes from small samples without taking into account the severity of the motor skill impairment. Poor motor coordination will hamper many ADL activities, keeping balance while putting on pants, close shirt buttons, lace shoes or drink from a cup without spilling. “Information about the impact of the severity of motor coordination level on the mastering these skills will improve the understanding of the disorder and help to define priorities for rehabilitation.”….

“It is therefore critical to identify children who lack musculoskeletal fitness prerequisites needed for sports activities at a young age. 

“”To make the picture complete, reports what parent see their children actually do during everyday life and the perceived competence for perspective of the child are needed in addition to standardized motor and fitness tests as that information will be the starting point for any intervention. “ “Although, low values for health-related fitness components, participation and perceived competency have been noted in the literature, studies having data on all these components in a large group of children with verified DCD criteria mentioned in the Diagnostic and Statistical Manual of mental disorders, fifth edition (DSM-5), over the full range of low motor scores and from a non-Western background are scarce. Little work has examined the consequences of the level of motor deficiency regarding these issues.

5. Line 92: ‘needed sports’ ‘needed for sports’ 

Response: 

Thanks. It has been changed on line 114

6. Methods: 

Line 122-123: The DCDDailyQ and CSAPPA were translated into the Arabic language. Was the backward-forward method applied? 

Response 

Yes, the backward-forward method was applied to translate the DCDDaily-Q and the CSAPPA into the Arabic language. The information has been added in the Study Design or Procedure of Method section on page 8 line 150.

7. Line 126: was the CSAPPA individually administered to the children? Please explain in the text Yes, the CSAPPA was individually administered to the children. 

Response 

We have added the explanation to the text in the Study Design or Procedure section on page 8 line 155-159.

8. Line 141: ‘parental questionnaire’. This questionnaire has not been mentioned before. What kind of questionnaire was used? 

Response 

Thank you, we have added the parental questionnaire’ and the reference: Psychometric properties of the DCDDaily-Q: a new parental questionnaire on children's performance in activities of daily living B. W. van der Linde, J. J. van Netten, B. E. Otten, K. Postema, R. H. Geuze and M. M. Schoemaker. Research in developmental disabilities 2014 Vol. 35 Issue 7 Pages 1711-1719 

The information has been added to the Participants section for Materials and Methods page 9 line 173.

9. Line 147: how many children were selected by teachers, and how many met the diagnostic criteria for DCD? 

Response 

In total 137 children were selected by teachers, and 109 children met the diagnostic criteria for DCD. The information has been added in the Participants section Materials and Methods on page 9 line 182-185. And a flowchart was added. (new figure 1)

10. DCDDailyQ and CSAPPA: please add that only Dutch/Canadian norms were available for these instruments. 

Response 

Thank you for pointing this out. The information has been now also been added in the method section on page 10, 11 line 226, 227, 234 and 237.

11. Line 198-199: “Non-parametric correlations were calculated for the TD and DCD groups, separately.” Please add between which variables correlations were calculated/ 

Response 

We added the variables Line 247, 248

Non-parametric correlations were calculated for activities of daily living (DCDDaily-Q), self-efficacy CSAPPA) and musculoskeletal fitness (PERF-FIT) the TD and DCD groups, separately. Significance was set at p< 0.05.

12. Considering the number of post-hoc tests: was a Bonferroni correction applied? 

Response 

Yes, both for the parametric and non-parametric testing. “ANCOVA with post hoc pairwise comparison with Bonferroni correction, was performed to investigate differences in musculoskeletal fitness between the 3 groups. ANCOVA (QUADE) with BMI as covariate and with correction for multiple testing”.

13. Results: 

Line 205: “The descriptive statistics revealed that sex was equally distributed between TD and DCD groups (p=0.275)”. Did you also check whether sex was equally distributed between the two DCD groups? At first glance, it seems that far more girls than boys were in the sDCD group compared to the mDCD group. 

Response 

The reviewer is right. We have almost equal number of boys and girls TD and DCD group. The stats in the paper refer to the 3 group comparison. The 2group comparison has a much higher p-value (p.0.59) We have adapted to text in the following way” The descriptive statistics revealed that sex was equally distributed between TD and the two DCD groups (p=0.275).”

14. Table 1: Height- please present two figures after the full stop. 

Response 

Two decimals have been added. 

15. Table 2: Please add measurement units for the PERF-FIT subtests. Please add in column 3 what the p-values represent (differences between TD and DCD groups?). I was confused by the label BMI, please explain. 

Response 

Thank you for pointing out this omission. The units have been added in Table 2 and that BMI values are for the covariate.

Because of the difference between groups in BMI, BMI was entered as covariate. In this column it is shown that the impact of BMI was significant on all power and agility items and also for throwing hopping/jumping. 

We have also added the classification for the BMI, which shows why using this covariate was necessary. (New Figure 2)

16. The DCDDailyQ is written in different ways throughout the text (DCD-DAILY-Q; DCDDaily-Q). In the papers by Van der Linde, the author of the questionnaire, it is called the DCDDaily-Q 

Response 

The name of the questionnaire was changed to reflect the way the author wrote it. 

17. Line 299: “Data also suggest that the CwDCD did not estimate their adequacy correct”. This is not what the data reveal. Children with CwDCD rated their adequacy comparable to the TD group. This does not have to mean that it is incorrect. Young children may not be able to reflect on their own performance. The same statement also occurs in the discussion. Please reflect on this 

Response 

The reviewer is right we have adapted the text:” Young children may not be able to reflect on their own performance”. 

18. Discussion: 

Lines 305-317: I miss a summary of the relation between the PERF-FIT and the questionnaires, where some interesting relations were revealed for the DCD group between participation/predilection and power/agility and motor skills. 

Response 

Thank you for pointing out this omission we have added text to discuss this finding,

” The relation between self-efficacy and DCDDaily-Q was different in the two DCD groups. The highest association with motor learning as reported by the parents was found with perceived enjoyment of physical education by the child in the moderate DCD group, if it took them longer to learn motor skills, they liked PE classes less. In the severe DCD group, the highest association with the time to learn skills emerged with perceived adequacy in physical education...”

“Remarkably, although low, the association of CSAPPA Predilection and total CSAPPA with PERF-FIT subscale was higher for power and agility than for the subscale Performance including catching, hopping and balance. Moreover, for the DCDDaily-Q participation the highest correlation was also found with Power and agility.”

19. Lines 318 and following: I miss a reference to the review of Rivilis (Rivilis, I, Hay J, Cairney, J, Klentrou, P, Liu J, Faught, BE. Physical activity and fitness in children with developmental coordination disorder: a systematic review. Res Dev Disabil. 2011, May-Jun; 32(3):894-910. Doi: 10.1013/j.ridd.2011.01.017. PMID: 21310588 

Response 

Thank you, we have added the reference 

20. Line 323: “Flexibility was assessed using the sit-and-reach test, which measures the flexibility of hamstrings and lower back, and not overall joint flexibility (26)”. This sentence should be added to the methods section, where information of the sit-and-reach is scarce. 

Response 

Line 208-215 We gave added the info to the method section. 

“The sit-and-reach test was added to assess flexibility and was assessed using a standardized wooden box. Participants are required to sit with knees straight and legs together and the soles of their feet against a box. They had to bend forward and reach as far as possible, with two hands on top of each other and hold the position for at least 3 s. The best score out of 3 trials (in cm) was further analyzed. The level of the feet is used as zero, so that if children do not reach their toes the values is negative and if the past their toes positive. The sit and reach test has the advantage of allowing for a simple estimation of flexibility of hamstrings and lower back in a short amount of time (26).”

21. Line 325: “We recommend using goniometry of the most important joints, to get valid information about the level of hyper- or hypomobility (27)”. This comment comes out of the blue, as hyper-or hypomobility is not discussed before. 

Response 

We have deleted this remark and adapted the sentence.

22. Line 327: What I miss in the discussion about BMI is the fact that the BMI levels of the children in the present study are still within the normal range. The text now reads as if the children with DCD are overweight, which is not the case. 

Response 

This is an important observation for us that the reviewer interpreted this sentence differently. Because in fact the children with s-DCD are overweight based of age and gender norms. We have added this info on page 13 added a figure to show the large difference between the groups.

23. Line 342: “Our data suggest that children’s motivation to participate in leisure sports is significantly influenced by their level of skill in motor performance”. This is to bold, as no causal directions were tested. Children with DCD indicated to be less motivated to participate in leisure sports. This may be due to their lower level of motor skill. 

Response 

We changed the text to : Our data suggest that children’s motivation to participate in leisure sports is significantly related to their level of skill in motor performance, which was also reported by Larsen et al. [39]. This is most likely due to their lower level of motor skill.

Line 345-347: “In fact, in both groups the highest correlation between predilection to physical activity and participation was seen for the fitness scale of the PERF-FIT and not for the motor skill item series”. Do you have an explanation? Maybe because predilection to physical activity is related to leisure sports, which require fitness? We have added our interpretation to the discussion 

“Remarkably, although low, the association of CSAPPA Predilection and total CSAPPA with PERF-FIT subscale was higher for Power and agility than for the subscale Performance including catching, hoping and balance items. This may be because predilection to physical activity is related to leisure sports, which require fitness. Moreover, for the DCDDaily-Q participation the highest correlation was also found with Power and agility. “…

“ This might be explained by findings in an earlier study that revealed the scores on the PERF-FIT are significantly related to performance in active play [42]. Outdoor play and sport-like games in children are characterized by short periods of intensive physical activity, interspersed with short periods of reduced or less intensive activity [43]. If children have lower levels of explosive muscle power, anaerobic capacity and agility, this will lead to restrictions in participation [44, 45].?

24. I miss a discussion about the differences between TD and DCD groups regarding ADL and CSAPPA data, and how they related to previous studies. You refer to Noordstar and Volman, but not in relation to your own findings 

Response 

We have added another study that used CSAPPA and one the used DCDDaily-Q . Not many studies have used the tools, besides in the respective psychometric papers.

25. Line 396:”This study showed that components of health-related fitness are not attainable for individuals with low motor competence”. I disagree. Your study shows that children with DCD have lower fitness levels, not that health-related fitness are not attainable. I do hope that these children can improve with training. 

Response 

We changed to text to according to your positive suggestion: “Children with DCD have lower fitness levels, hopefully with targeted intervention sufficient levels of health-related fitness are attainable”.

26. I miss in the discussion a remark about the different boys-girls ratios in the mDCD and sDCD groups and how this may (or not) have affected your results 

Response 

See above No significant differences were found in the frequency of sex over the 3 groups.

Reviewer 2

1. “Activities of daily living, self-efficacy and musculoskeletal fitness are compromised in children with developmental coordination disorder” may be changed to “Activities of daily, living, self-efficacy and musculoskeletal comparison among the children with developmental coordination disorder” 

Response 

Based on suggestion of the reviewer we have changed to title to emphasize the uniqueness of our study. Activities of daily living, self-efficacy and musculoskeletal fitness and the interrelation in children with moderate and severe Developmental Coordination Disorder

2.Please refer to: “In total 214 children participated in the study: 105 TDC and 109 CwDCD”. How the sample size was determined is unclear. Sample size selection may be verified using Lwanga and Leneshow (1990). Lwanga, S.K., Lemeshow, S. and World Health Organization, 1991. Sample size determination in health studies: a practical manual. World Health Organization 

Response 

Given that we were interested in medium (>50) to large effect sizes of the difference between TD and DCD and taking into account that both parametric and non-parametric testing would be needed we aimed at safe numbers to find the differences between TD and DCD. With these numbers we were also still likely to find differences in case of the analysis with 3 groups if effects were large. We have added the power calculation to the paper.

3. Author may provide flow diagram research methodology steps for sample recruitment and follow up for more clarity 

Response 

Thank you for this suggestion.

A flow diagram of the research methodological steps for sample recruitment has been added for more clarity 

Response 

as suggested by the Reviewer; Fig 1: Selection of children and tests administered, has been added to the Participants section.

4. The legend shown below Table 1 may be changed, M-DCD may be changed to m-DCD, similarly S-DCD may be changed to s-DCD to make uniform throughout the manuscript 

Response 

The legends were changed as suggested p.11- Table 1

5. Table 2 may be properly formatted to identify the heading of Power and Agility, Motor item series, and Flexibility 

Response 

The headings were written in bold to identify them clearly p.12- Table 2

6. In Table 2 the heading F may be F-value, legend and last column entry “all sign” may be “all significant” since “all sign” has misleading meaning, similarly “TD-DCD sign” may be “TD-DCD significant”. 

Response 

The table entries were changed as suggested p.12- Table 2

7. Table 3 first column suitable heading may be provided for instance. Test description. The dF2,211 may be added at the table bottom. The degree of freedom (DF) may be used throughout the manuscript. 

Response 

We have added the headings and taken the Df out

8. Table 4 first two columns may be suitably named as DCD types and Statistical parameters respectively 

Response 

We have added info.

9. Notes below the Tables may be suitably changed to small font size or maybe in Italics. All notes must be below adjoining the table. 

Response 

The notes below the tables were changed to a smaller font size 

10. Some typos may be checked for instance “table 3” should be “Table 3”, “Table 4, gives…” should be “Table 4, gives…” and “predilection of the children (Table 5)” should be “predilection of the children (please refer to Table 5).” “by Larsen et al. (29)” should be “by Larsen et al. (29).” 

Response 

The corrections were made as suggested p.15 Line 297

p.16 Line 322

p.22 Line 413

11. Table header for Table 5 may be suitably changed. The values of Spearman’s rho are shown in the last three columns. The first two columns must be suitably named. 

Response 

Thank you for this suggestion it has been changed accordingly

12. Table header for first two columns of Table 6 may be suitably added. 

Response 

Headers have been added

13. Asper Jelsma et al. (2013), “In general the motor performance and joint mobility are not related in DCD”. Hence recommending the use of goiniometry may be disjoint here. 

Jelsma, L.D., Geuze, R.H., Klerks, M.H., Niemeijer, A.S. and Smits-Engelsman, B.C. 2013. The relationship between joint mobility and motor performance in children with and without the diagnosis of developmental coordination disorder, BMC pediatrics, 13, pp.1-8. 

Response 

We have taken this out and adapted the sentence. Added info from the Jelsma paper.

“We found lower flexibility in the CwDCD, while Jelsma et al. [36] reported higher prevalence of hypermobility in the DCD-group. Most likely this is caused by the different outcomes used; Beighton score versus Sit and Reach. To get valid information about the level of hyper- or hypomobility it is recommended to measure range of motion of the most important joints [37]. “

14. Some recommendations put forward by Mandich et al., (2001) as “Skill acquisition through evidence-based practices” Interventions leading to functional outcomes etc. may also be discussed

Mandich, A.D., Polatajko, H.J., Macnab, J.J. and Miller, L.T., 2001. Treatment of children with developmental coordination disorder: What is the evidence? Physical & occupational therapy in pediatrics, 20(2-3), pp.51-68. 

Response 

Thank you for these suggestions.

We decided that explicit discussion of different forms of intervention was outside the scope of this paper. We therefore referred to the international DCD guideline, in which more recent studies are summarized and rated for their level of evidence.

15. Some similar studies related to PERF-FIT, DCD may be compared with the present studies:

• Smits-Engelsman, B., Neto, J.L.C., Draghi, T.T.G., Rohr, L.A. and Jelsma, D., 2020. Construct validity of the PERF-FIT, a test of motor-skill-related fitness for children in low resource areas. Research in developmental disabilities, 102, p.103663

• Girish S, Raja K, Kamath A. Prevalence of developmental coordination disorder among mainstream school children in India. Journal of pediatric rehabilitation medicine. 2016 Jan 1:9(2):107-16.

• Van der Linde, B.W., Van Netten, J.J., Otten, B., Postema, K., Geuze, R.H., and Schoemaker, M.M., 2013. Development and psychometric properties of the DCDDaily: a new test for clinical assessment of capacity in activities of daily living in children with developmental coordination disorder. Clinical Rehabilitation, 27(9), pp.834-844. 

Response 

Thank you for pointing out some interesting references. I have read the Girish S, paper with interest but couldn’t find the relation with our study, since it is a very well performed study into the prevalence of DCD, but comparison between different tools or different subgroups are not made. We had already added info from the two Van der Linde studies in the paper and have added a sentence about a comparison to the Brazilian study. “In line with the finding of Smits-Engelsman et al., in a group of Brazilian children, the present study confirmed how important it is for clinicians to follow up on skill-related fitness to address potential future adverse health effects.”

16. As authors all relevant data are within the manuscript and its supporting information files. However, no data file is uploaded. Authors also need to upload DCD Daily-Questionnaire and Children’s Self-perceptions of Adequacy in and Predilection for Physical Activity Questionnaire 

Response 

I don’t think we should upload questionnaires that are not ours, only the data we collected with those tools. We had permission to use the CSAPPA tool but not to publish them. 

The DCDdaily_Q is available online 

http://dcddaily.com/assets/manual_dcddaily-q_feb2018.pdf

---

## [Decision Letter · Decision Letter 1]

27 Nov 2023

PONE-D-23-30683R1Activities of daily living, self-efficacy and musculoskeletal fitness and the interrelation in children with moderate and severe Developmental Coordination DisorderPLOS ONE

Dear Dr. Farhat,

Thank you for submitting your manuscript to PLOS ONE. After careful consideration, we feel that it has merit but does not fully meet PLOS ONE’s publication criteria as it currently stands. Therefore, we invite you to submit a revised version of the manuscript that addresses the points raised during the review process.

We look forward to receiving your revised manuscript.

Kind regards,

Mohamed Rafik N. Qureshi, Ph.D.

Academic Editor

PLOS ONE

Journal Requirements:

Additional Editor Comments:

The manuscript entitled "Activities of daily living, self-efficacy and musculoskeletal fitness and the interrelation in children with moderate and severe Developmental Coordination Disorder" has been modified as per the reviewer's suggestions. The manuscript may further be revised as per the reviewer's remarks. The authors have not made all the data available, except a 'Group Comparison TD DCD PLos one.sav'.

Reviewers' comments:

Reviewer's Responses to Questions

**Comments to the Author**

1. If the authors have adequately addressed your comments raised in a previous round of review and you feel that this manuscript is now acceptable for publication, you may indicate that here to bypass the “Comments to the Author” section, enter your conflict of interest statement in the “Confidential to Editor” section, and submit your "Accept" recommendation.

Reviewer #1: (No Response)

Reviewer #2: All comments have been addressed

2. Is the manuscript technically sound, and do the data support the conclusions?

Reviewer #1: Yes

Reviewer #2: Yes

3. Has the statistical analysis been performed appropriately and rigorously? 

Reviewer #1: Yes

Reviewer #2: Yes

4. Have the authors made all data underlying the findings in their manuscript fully available?

Reviewer #1: Yes

Reviewer #2: No

5. Is the manuscript presented in an intelligible fashion and written in standard English?

Reviewer #1: No

Reviewer #2: Yes

6. Review Comments to the Author

Reviewer #1: The authors responded in a satisfactory way to my comments. Some typo's/ clarifications are still needed:

Abstract:

Slow motor learning was associated with lower perceived enjoyment in physical education in the moderate DCD group, and with lower perceived adequacy in physical education in the severe DCD group.

Children with DCD participate and enjoy physical activity less than their peers. This combination of lower participation, lower predilection to physical activity ….

Introduction:

Line 87: “Our current study extends on Cairney’s work [15], who reported that children with DCD had lower levels of generalized than TD children. “ A word is missing after generalized.

Methods:

Line 182: “In total 137 children were selected by teachers, and 141 children met the diagnostic criteria for DCD.” Typo? 141 is more than 137.

Line 213: “The level of the feet is used as zero, so that if children do not reach their toes the values is negative and if the past their toes positive.” Change into: the value is negative; if they reach past their toes.

Line 226: “Norms are available based a Dutch and Spanish sample.” Change into: based upon

Line 248: “Non-parametric correlations were calculated for activities of daily living (DCDDaily-Q), self-efficacy CSAPPA) and musculoskeletal fitness (PERF-FIT) the TD and DCD groups, separately.” Change into: were calculated between

Line 264: “Fig 2. Percentage per the BMI classification” Change into: Percentage per BMI classification.

Discussion:

Line 416:” In the severe DCD group, the highest association with the time to learn skills emerged with perceived adequacy in physical education” Please add: the longer it took them to learn motor skills, the lower their perceived adequacy in physical education.

Line 467:” Moreover, for the DCDDaily-Q participation the highest correlation was also found with Power and agility.” Please add direction: children who participated more had better scores for power and agility?

Reviewer #2: 1) ‘Materials and Methods Study Design or Procedure’ may be changed to under section ‘Materials and Methods Study Design Procedure’

2) There are some typos may be corrected for instance:

a) components; should be components:

b) ‘not feel adequate in physical fitness tasks, such’ should be ‘not feel adequate for physical fitness tasks such’

c) ‘who lack musculoskeletal fitness prerequisites needed sports should be ‘who lack the musculoskeletal fitness prerequisites needed for sports’

d) To make the picture complete, reports what parent see should be To make the picture complete, reports of what parents see

e) ‘ life and perspective of the child’ should be ‘life and the perspective of the child’

f) ‘Diagnostic and Statistical Manual of mental disorders,’ should be ‘Diagnostic and Statistical Manual of Mental Disorders,’

g) ‘Sit-and-reach test was added as measure of flexibility.’ should be ‘A sit-and-reach test was added as a measure of flexibility.’

7. PLOS authors have the option to publish the peer review history of their article (what does this mean?). If published, this will include your full peer review and any attached files.

Reviewer #1: No

Reviewer #2: No

---

## [Author Response · Author response to Decision Letter 1]

20 Dec 2023

Reviewer 1

We thank the reviewer for her valuable and detailed comments which helped to improve the paper. 

Abstract:

Slow motor learning was associated with lower perceived enjoyment in physical education in the moderate DCD group, and with lower perceived adequacy in physical education in the severe DCD group.

Children with DCD participate and enjoy physical activity less than their peers. This combination of lower participation, lower predilection to physical activity …. 

Responses:

Thanks. We have changed the sentence: “Slow motor learning was associated with perceived enjoyment of physical education in the moderate DCD group, and with perceived adequacy in physical education in the severe DCD group.” By “Slow motor learning was associated with lower perceived enjoyment in physical education in the moderate DCD group, and with lower perceived adequacy in physical education in the severe DCD group.”

Introduction:

Line 87: “Our current study extends on Cairney’s work [15], who reported that children with DCD had lower levels of generalized than TD children. “ A word is missing after generalized. 

Responses:

Thanks. We have added the words missing: “self-efficacy regarding physical activity” and changed the sentence: “Our current study extends on Cairney’s work [15], who reported that children with DCD had lower levels of generalized self-efficacy regarding physical activity than TD children.”

Methods: 

Line 182: “In total 137 children were selected by teachers, and 141 children met the diagnostic criteria for DCD.” Typo? 141 is more than 137. 

Responses:

Thanks. We have corrected the missing: “In total 167 children were selected by teachers, and 141 children met the diagnostic criteria for DCD.”

Line 213: “The level of the feet is used as zero, so that if children do not reach their toes the values is negative and if the past their toes positive.” Change into: the value is negative; if they reach past their toes. 

Responses:

Thank you, we have changed the sentence as suggested by the reviewer: “The level of the feet is used as zero, the value is negative; if they reach past their toes.”

Line 226: “Norms are available based a Dutch and Spanish sample.” Change into: based upon 

Responses:

Thank you, we have changed the sentence: “Norms are available based a Dutch and Spanish sample.” By “Norms are available based upon Dutch and Spanish sample.”

Line 248: “Non-parametric correlations were calculated for activities of daily living (DCDDaily-Q), self-efficacy CSAPPA) and musculoskeletal fitness (PERF-FIT) the TD and DCD groups, separately.” Change into: were calculated between 

Responses:

Thank you, we have changed the sentence: “Non-parametric correlations were calculated for activities of daily living (DCDDaily-Q), self-efficacy CSAPPA) and musculoskeletal fitness (PERF-FIT) the TD and DCD groups, separately.” By “Non-parametric correlations were calculated between activities of daily living (DCDDaily-Q), self-efficacy CSAPPA) and musculoskeletal fitness (PERF-FIT) the TD and DCD groups, separately.”

Line 264: “Fig 2. Percentage per the BMI classification” Change into: Percentage per BMI classification. 

Responses:

Thanks. We have changed the sentence: “Fig 2. Percentage per the BMI classification” by “Fig 2. Percentage per BMI classification.”

Discussion: 

Line 416:” In the severe DCD group, the highest association with the time to learn skills emerged with perceived adequacy in physical education” Please add: the longer it took them to learn motor skills, the lower their perceived adequacy in physical education. 

Responses:

Thanks. We have changed the sentence: “In the severe DCD group, the highest association with the time to learn skills emerged with perceived adequacy in physical education.” by “In the severe DCD group, the highest association with the longer it took them to learn motor skills, the lower their perceived adequacy in physical education.” As suggested by the reviewer.

Line 467:” Moreover, for the DCDDaily-Q participation the highest correlation was also found with Power and agility.” Please add direction: children who participated more had better scores for power and agility? 

Responses:

Based on a suggestion of the reviewer we have added the sentence: “Indeed, children who participated more in physical activities had better ability to change body velocity and direction.”

Reviewer 2

1) ‘Materials and Methods Study Design or Procedure’ may be changed to under section ‘Materials and Methods Study Design Procedure’ 

Responses:

Based on the suggestion of the reviewer we have changed “Study Design or Procedure” to “Study Design Procedure”. Line 144.

2) There are some typos may be corrected for instance:

a) components; should be components: 

Responses:

Thank you for this suggestion. Line 73.

We have changed “components;” to “components:”

b) ‘not feel adequate in physical fitness tasks, such’ should be ‘not feel adequate for physical fitness tasks such’ 

Responses:

Thanks. We have changed “not feel adequate in physical fitness tasks, such” to “not feel adequate for physical fitness tasks such”. Line 79.

c) ‘who lack musculoskeletal fitness prerequisites needed sports should be ‘who lack the musculoskeletal fitness prerequisites needed for sports’ 

Responses:

Thanks. We have changed the sentence: “who lack musculoskeletal fitness prerequisites needed sports” by “who lack the musculoskeletal fitness prerequisites needed for sports”. Line 115.

d) To make the picture complete, reports what parent see should be To make the picture complete, reports of what parents see 

Responses:

The corrections were made as suggested in Line 116.

We have changed: “To make the picture complete, reports what parent see” by “To make the picture complete, reports of what parents see”

e) ‘ life and perspective of the child’ should be ‘life and the perspective of the child’ 

Responses:

Thank you for this suggestion it has been changed accordingly in line 117.

f) ‘Diagnostic and Statistical Manual of mental disorders,’ should be ‘Diagnostic and Statistical Manual of Mental Disorders” 

Responses:

Thank you for this suggestion it has been changed accordingly in lines 122-123.

g) ‘Sit-and-reach test was added as measure of flexibility.’ should be ‘A sit-and-reach test was added as a measure of flexibility.’ 

Responses:

Thank you for these suggestions. Line 209.

We have changed “Sit-and-reach test was added as measure of flexibility.” by “A sit-and-reach test was added as a measure of flexibility.”

---

## [Decision Letter · Decision Letter 2]

12 Jan 2024

PONE-D-23-30683R2Activities of daily living, self-efficacy and musculoskeletal fitness and the interrelation in children with moderate and severe Developmental Coordination DisorderPLOS ONE

Dear Dr. Farhat,

Thank you for submitting your manuscript to PLOS ONE. After careful consideration, we feel that it has merit but does not fully meet PLOS ONE’s publication criteria as it currently stands. Therefore, we invite you to submit a revised version of the manuscript that addresses the points raised during the review process.Please submit your revised manuscript by Feb 26 2024 11:59PM. If you will need more time than this to complete your revisions, please reply to this message or contact the journal office at plosone@plos.org. Please include the following items when submitting your revised manuscript:A rebuttal letter that responds to each point raised by the academic editor and reviewer(s). You should upload this letter as a separate file labeled 'Response to Reviewers'.A marked-up copy of your manuscript that highlights changes made to the original version. You should upload this as a separate file labeled 'Revised Manuscript with Track Changes'.An unmarked version of your revised paper without tracked changes. You should upload this as a separate file labeled 'Manuscript'.If applicable, we recommend that you deposit your laboratory protocols in protocols.io to enhance the reproducibility of your results. Protocols.io assigns your protocol its own identifier (DOI) so that it can be cited independently in the future. For instructions see: https://journals.plos.org/plosone/s/submission-guidelines#loc-laboratory-protocols. Additionally, PLOS ONE offers an option for publishing peer-reviewed Lab Protocol articles, which describe protocols hosted on protocols.io. Read more information on sharing protocols at https://plos.org/protocols?utm_medium=editorial-email&utm_source=authorletters&utm_campaign=protocols.

We look forward to receiving your revised manuscript.

Kind regards,

Mohamed Rafik N. Qureshi, Ph.D.

Academic Editor

PLOS ONE

Journal Requirements:

Additional Editor Comments :

The manuscript entitled"Activities of daily living, self-efficacy and musculoskeletal fitness and the interrelation in children with moderate and severe Developmental Coordination Disorder

' needs modification as per the reviewers' comments.

Reviewers' comments:

Reviewer's Responses to Questions

**Comments to the Author**

1. If the authors have adequately addressed your comments raised in a previous round of review and you feel that this manuscript is now acceptable for publication, you may indicate that here to bypass the “Comments to the Author” section, enter your conflict of interest statement in the “Confidential to Editor” section, and submit your "Accept" recommendation.

Reviewer #2: (No Response)

Reviewer #3: All comments have been addressed

2. Is the manuscript technically sound, and do the data support the conclusions?

Reviewer #2: Yes

Reviewer #3: Partly

3. Has the statistical analysis been performed appropriately and rigorously? 

Reviewer #2: Yes

Reviewer #3: Yes

4. Have the authors made all data underlying the findings in their manuscript fully available?

Reviewer #2: (No Response)

Reviewer #3: Yes

5. Is the manuscript presented in an intelligible fashion and written in standard English?

Reviewer #2: (No Response)

Reviewer #3: No

6. Review Comments to the Author

Reviewer #2: Authors have modified the manuscript as per the reviewers' comments.

However, the manuscript is not free from typos, hence needs careful reading and editing. Some of the typos are listed below:

'To determine musculoskeletal fitness level ..' should be 'To determine musculoskeletal fitness levels ..'

'Lastly, the relation physical fitness, ..' should be 'Lastly, the relation of physical fitness, ..'

'Key words' should be 'Keywords'

'By definition, CwDCD have ..' should be 'By definition, CwDCD has ..'

'CwDCD face evident motor difficulties ..' should be 'CwDCD faces evident motor difficulties ..'

'on the mastering these skills.. ' should be 'on mastering these skills... '

'...that CwDCD participate less in physical activities ...' should be '...that CwDCD participates less in physical activities ...'

'...on their observation on the playground...' should be '...on their observations on the playground...'

'...they did not estimate their adequacy correct. ' should be 't....hey did not estimate their adequacy correctly. '

Reviewer #3: OVERALL

•The overall review of the document is positive, highlighting its elaborate and well-conducted nature. The depth of the study design, ethical considerations, recruitment process, translation methods, and assessment procedures is commendable.

GENERAL

•The study's objective focuses on musculoskeletal fitness in DCD and its association with the level of ADL perceived by parents and self-efficacy as perceived by children. To enhance clarity and better align with the study's objective, it is recommended to consistently use the term "musculoskeletal fitness" throughout the document. This will help avoid potential ambiguity associated with broader terms like "health-related fitness" and "motor skill-related fitness."

•Additionally, consider providing a clear operational definition of the term "musculoskeletal fitness" within the document. This adjustment ensures that the terminology aligns precisely with the study's focus and minimizes the risk of confusion among readers.

•Throughout the entire document, there is room for improvement in the use of English

SPECIFIC

1.INTRODUCTION

The introduction offers a thorough background on DCD. To enhance its effectiveness and improve reader comprehension, it is suggested to refine the introduction emphasizing on particular aspects of musculoskeletal fitness and its correlation with activities of daily living (ADL) and self-efficacy, coupled with a more explicit differentiation from prior studies

2.MATERIALS & METHODS

•Study design and procedure: Provides a clear understanding of the ethical considerations, recruitment process, translation methods, and assessment procedures

•Participants:

oThe statement regarding the identification of children with a motor coordination problem by parents or teachers (Criterion B) does not explicitly align with the DSM-5 criteria for the diagnosis of DCD. To strengthen this aspect, consider revising the statement to explicitly state how the motor coordination difficulties identified by parents or teachers, as per Criterion B, specifically impact activities of daily living or academic performance. This clarification will ensure a closer alignment with the diagnostic criteria for DCD.

oIf the acerating criteria D is solely reliant on parent interviews without involvement of specialist, it is essential to acknowledge this as a limitation of the study. Additionally discuss how this limitation might impact the overall validity.

oIf the global prevalence of DCD is estimated to be 5%, & the study identified 141 out of 167 with DCD, it does indeed raise questions about the potential overestimation. In light of this discrepancy, it is crucial for the study to thoroughly discuss and address the reasons behind the higher-than-expected prevalence in the sample. Consider examining the sensitivity and specificity of the diagnostic tools used, potential biases in the identification process, or any other factors that might have contributed to the observed prevalence. Additionally, discuss how this overestimation might impact the generalizability of the study's findings and the validity of the prevalence rates reported.

•Measurement: The measurements in the study are indeed well-explained, and their psychometric properties are clearly demonstrated. However, the evaluation of cultural sensitivity, particularly pertaining to the items in the Movement Assessment Battery for Children, Second Edition (MABC-2), is not explicitly addressed. It would be beneficial for the study to include a discussion or statement regarding the cultural sensitivity of MABC-2 items

•Statistical analysis: While all the tests were deemed suitable, there is a need for clarification regarding the particular non-parametric correlation method applied in the study concerning DCDDaily-Q, CSAPPA, and PERF-FIT.

•Results: The results have been comprehensively and effectively documented. However, there is a need for clarification regarding the specific non-parametric test used in Table 5

•Discussion: The document features a comprehensive discussion. However, potential confusion among readers may arise due to the use of terms like "health-related" and "musculoskeletal" fitness. Therefore, it is recommended to revise the wording for clarity, and addressing the limitation of the study is advised

7. PLOS authors have the option to publish the peer review history of their article (what does this mean?). If published, this will include your full peer review and any attached files.

Reviewer #2: No

Reviewer #3: **Yes: **Srilatha Girish

---

## [Author Response · Author response to Decision Letter 2]

31 Jan 2024

Reviewer 2 

We thank the reviewer for her valuable and detailed comments which helped to improve the paper. 

1) 'To determine musculoskeletal fitness level ..' should be 'To determine musculoskeletal fitness levels ..'

Response

Thanks. We have changed “level” By “levels” on page 2, line 22..

2) 'Lastly, the relation physical fitness, ..' should be 'Lastly, the relation of physical fitness, ..'

Response 

Thanks. We have changed the sentence: “'Lastly, the relation physical fitness, .” By “'Lastly, the relation of physical fitness” in page 2, line 25.

3) 'Key words' should be 'Keywords' 

Response 

Thanks. We have changed “Key words” By “'Keywords”

4) 'By definition, CwDCD have ..' should be 'By definition, CwDCD has ..' Because CwDCD stance for children with DCD, this a plural. 

Response 

To avoid confusion, we have written it full out throughout the text. 

5) 'CwDCD face evident motor difficulties ..' should be 'CwDCD faces evident motor difficulties Response

CwDCD stance for children with DCD this a plural. To avoid confusion, we have written it full out throughout the text.

6) 'on the mastering these skills.. ' should be 'on mastering these skills... ' 

Response 

Because CwDCD stance for children with DCD this a plural. To avoid confusion, we have written it full out throughout the text.

7) '...that CwDCD participate less in physical activities ...' should be '...that CwDCD participates less in physical activities ...' 

Response 

Because CwDCD stance for children with DCD this a plural. To avoid confusion, we have written it full out throughout the text.

8) '...on their observation on the playground...' should be '...on their observations on the playground...' 

Response 

Thanks. We have changed the sentence: “on their observation on the playground” By “on their observations on the playground” in page 9, line 182. 

9) '...they did not estimate their adequacy correct. ' should be 't....hey did not estimate their adequacy correctly. 

Response 

Thanks. We have changed the sentence: “...they did not estimate their adequacy correct” By “they did not estimate their adequacy correctly ” in page 19 line 388.

Reviewer 3

1) GENERAL 

The study's objective focuses on musculoskeletal fitness in DCD and its association with the level of ADL perceived by parents and self-efficacy as perceived by children. To enhance clarity and better align with the study's objective, it is recommended to consistently use the term "musculoskeletal fitness" throughout the document. This will help avoid potential ambiguity associated with broader terms like "health-related fitness" and "motor skill-related fitness." 

Response 

Thank you for pointing this out. We have added the definitions of the used terms to the introduction and have consistently used those terms throughout the document as suggested by the reviewer. We choose to use definitions provided by 

Corbin, C.B., Pangrazi, R.P. and Franks, B.D. (2000) Definition: Health, Fitness and Physical Activity. President’s Council on Physical Fitness and Sports Research Digest, 3, 1-8. The two types of physical fitness most often identified are health-related physical fitness and motor skill-related physical fitness. 

Health-related physical fitness consists of those components of physical fitness that have a relationship with good health. The components are commonly defined as body composition, cardiovascular fitness, flexibility, muscular endurance, and strength. Motor skill-related fitness incorporates agility, balance, coordination, speed, power, and reaction time, reflecting the performance aspect of physical fitness [11]. Because we use the PERF-FIT as primary outcome, which was specifically developed as is a valid test to measure motor skill-related fitness fitness , we prefer to stick to that term (and not musculoskeletal fitness). Motor skill-related fitness incorporates motor potential to carry out physical activity. 

2) Additionally, consider providing a clear operational definition of the term "musculoskeletal fitness" within the document. This adjustment ensures that the terminology aligns precisely with the study's focus and minimizes the risk of confusion among readers. 

Response 

Thanks. As suggested by the reviewer, we have added a definition of the different fitness terms to the introduction section on page 4, line 71-79: . Both motor skill competency and fitness are critical for participation in active play [7]. Compared to typically developing (TD) children, children with DCD are particularly likely to exhibit reduced fitness [9, 10] The two types of physical fitness most often identified are health-related physical fitness and motor skill-related physical fitness [10]. Health-related physical fitness consists of those components of physical fitness that have a relationship with good health. The components are commonly defined as body composition, cardiovascular fitness, flexibility, muscular endurance, and strength. Motor skill-related fitness incorporates agility, balance, coordination, speed, power, and reaction time, reflecting the performance aspect of physical fitness (Caspersen, Powell, & Christenson, 1985 ). 

3) Throughout the entire document, there is room for improvement in the use of English 

Response 

The manuscript has been checked our English professor and we hope this has improved the text. 

SPECIFIC 

1.INTRODUCTION 

4) The introduction offers a thorough background on DCD. To enhance its effectiveness and improve reader comprehension, it is suggested to refine the introduction emphasizing on particular aspects of musculoskeletal fitness and its correlation with activities of daily living (ADL) and self-efficacy, coupled with a more explicit differentiation from prior studies.

Response

Thank you for this suggestion. We have added the following text parts: 

“Children with DCD experience considerable difficulties controlling their body movements during functional motor tasks [7]. In some tests, like for anaerobic muscle endurance (e.g. the sit- or push-ups executed in 30 s the poor coordination of repetitive movements of the tasks as well as perceptions of poor physical fitness can have a negative impact on performance”. In page 5, line 88-92.

“Importantly, Ferguson et al. [12] showed that tasks that should have been learned implicitly during play and everyday activities (running up and down stairs, lifting a box, sit to stand from a chair) were consistently poor in children with DCD. Additionally, adequate muscle strength and endurance are important for performing many of these daily activities and to participate in sports without early fatigue [21]. “ In page 6, line 117-121.

Also, in the last full paragraph of the introduction we point out the uniqueness of out study 

“Although, low values of motor skill-related fitness components, participation and perceived competency have been noted in the literature, studies having data on all these components in a large group of children with verified DCD criteria mentioned in the Diagnostic and Statistical Manual of Mental Disorders, fifth edition (DSM-5), over the full range of low motor scores and from a non-Western background are scarce. Little work has examined the consequences of the level of motor deficiency regarding these issues. Therefore, we included two groups of children with DCD, one with significant movement difficulties and one at risk for motor difficulties. In page 6-7, line 132-139.

2. MATERIALS & METHODS 

5) •Participants: 

The statement regarding the identification of children with a motor coordination problem by parents or teachers (Criterion B) does not explicitly align with the DSM-5 criteria for the diagnosis of DCD. To strengthen this aspect, consider revising the statement to explicitly state how the motor coordination difficulties identified by parents or teachers, as per Criterion B, specifically impact activities of daily living or academic performance. This clarification will ensure a closer alignment with the diagnostic criteria for DCD. 

Response 

Thank you for these suggestions. We have added a sentence to clarify Criterion B: “the DCDDaily-Q was used to further assess interference of the motor impairment with daily activities and/or academic achievement.” Page 9, line 184-185.

6) If the acerating criteria D is solely reliant on parent interviews without involvement of specialist, it is essential to acknowledge this as a limitation of the study. Additionally, discuss how this limitation might impact the overall validity. 

Response 

We have added this recommendation to limitation section, page 26, line 552-556.

“ Moreover, concerning criteria D, intellectual or cognitive impairment, was evaluated by interviews with parents or teachers and not measured with an IQ-test. Neither were children seen by a neurologist to rule out other medical conditions that could explain impaired motor development. Thus, it might be possible that some children had unnoticed cognitive impairment or co-morbidities.”

7) If the global prevalence of DCD is estimated to be 5%, & the study identified 141 out of 167 with DCD, it does indeed raise questions about the potential overestimation. In light of this discrepancy, it is crucial for the study to thoroughly discuss and address the reasons behind the higher-than-expected prevalence in the sample. Consider examining the sensitivity and specificity of the diagnostic tools used, potential biases in the identification process, or any other factors that might have contributed to the observed prevalence. Additionally, discuss how this overestimation might impact the generalizability of the study's findings and the validity of the prevalence rates reported

Response 

Thank you for pointing this out because it gave the wrong impression. We have rewritten the selection of participants for DCD and TD children to eliminate any confusion (page 9, lines 199-203.). The prevalence of DCD estimated in the current study was 6.5 %. We have changed the sentence: “In total 167 children were selected by teachers, and 141 children met the diagnostic criteria for DCD. Finely, 214 children performed all the tests, and participated in the study: 105 TD and 109 CwDCD” into “In total 141 children were identified who met the diagnostic criteria for DCD out of 2170 children from the 6 elementary schools. TD children were matched according to age and sex to the children with DCD. Finely, 214 children performed all the tests, and participated in the study: 109 children with DCD and 105 typically developing matched controls. 

8) Statistical analysis: While all the tests were deemed suitable, there is a need for clarification regarding the particular non-parametric correlation method applied in the study concerning DCDDaily-Q, CSAPPA, and PERF-FIT. 

Response 

We have added that we used Spearman correlations between DCDDaily-Q, CSAPPA, and PERF-FIT. Page 12, line 270.

9) Discussion: The document features a comprehensive discussion. However, potential confusion among readers may arise due to the use of terms like "health-related" and "musculoskeletal" fitness. Therefore, it is recommended to revise the wording for clarity, and addressing the limitation of the study is advised 

Response 

We have revised the wording for clarity and addressed the limitation of the study as suggested by the reviewer.

We discussed parent interviews without the involvement of a school psychologist and that a neurologic examination to rule out other medical conditions was not performed (criteria D). We also added the cultural sensitivity of MABC-2 items in the limitations section (page 26, line 547-551).

---

## [Decision Letter · Decision Letter 3]

14 Feb 2024

Activities of daily living, self-efficacy and motor skill related fitness and the interrelation in children with moderate and severe Developmental Coordination Disorder

PONE-D-23-30683R3

Dear Dr. Farhat,

We’re pleased to inform you that your manuscript has been judged scientifically suitable for publication and will be formally accepted for publication once it meets all outstanding technical requirements.

Kind regards,

Mohamed Rafik N. Qureshi, Ph.D.

Academic Editor

PLOS ONE

Additional Editor Comments (optional):

Thank you for the updated version.

Reviewers' comments:

Reviewer's Responses to Questions

**Comments to the Author**

1. If the authors have adequately addressed your comments raised in a previous round of review and you feel that this manuscript is now acceptable for publication, you may indicate that here to bypass the “Comments to the Author” section, enter your conflict of interest statement in the “Confidential to Editor” section, and submit your "Accept" recommendation.

Reviewer #2: All comments have been addressed

Reviewer #3: All comments have been addressed

2. Is the manuscript technically sound, and do the data support the conclusions?

Reviewer #2: Yes

Reviewer #3: Yes

3. Has the statistical analysis been performed appropriately and rigorously? 

Reviewer #2: Yes

Reviewer #3: Yes

4. Have the authors made all data underlying the findings in their manuscript fully available?

Reviewer #2: Yes

Reviewer #3: Yes

5. Is the manuscript presented in an intelligible fashion and written in standard English?

Reviewer #2: (No Response)

Reviewer #3: Yes

6. Review Comments to the Author

Reviewer #2: The paper titled "Activities of daily living, self-efficacy and motor skill related fitness and the interrelation in children with moderate and severe Developmental Coordination Disorder" has been amended as per the comments.

Reviewer #3: (No Response)

7. PLOS authors have the option to publish the peer review history of their article (what does this mean?). If published, this will include your full peer review and any attached files.

Reviewer #2: No

Reviewer #3: **Yes: **Srilatha Girish

---

## [Editor Report · Acceptance letter]

5 Mar 2024

PONE-D-23-30683R3 

PLOS ONE

Dear Dr. Farhat, 

I'm pleased to inform you that your manuscript has been deemed suitable for publication in PLOS ONE. Congratulations! Your manuscript is now being handed over to our production team.

Kind regards, 

on behalf of

Prof.(Dr.) Mohamed Rafik N. Qureshi 

Academic Editor

PLOS ONE